# dLLM-Cache:
# Accelerating Diffusion Large Language Models with Adaptive Caching

Zhiyuan Liu [* 1 2]   Yicun Yang [* 1 2]   Yaojie Zhang [2 3]   Junjie Chen [2]   Chang Zou [1 2 3]   Qingyan Wei [2]
Shaobo Wang [1 2]   Yichen Zhu [4]   Linfeng Zhang [1 2]

## Abstract

Autoregressive Models (ARMs) have long dominated the landscape of Large Language Models. Recently, a new paradigm has emerged in the form of diffusion-based Large Language Models (dLLMs), which generate text by iteratively denoising masked segments. This approach has shown significant advantages and potential. However, dLLMs suffer from high inference latency. Traditional ARM acceleration techniques, such as Key-Value caching, are incompatible with dLLMs due to their bidirectional attention mechanism. To address this specific challenge, our work begins with a key observation that dLLM inference involves a static prompt and a partially dynamic response, where most tokens remain stable across adjacent denoising steps. Based on this, we propose dLLM-Cache, a training-free adaptive caching framework that combines long-interval prompt caching with partial response updates guided by feature similarity. This design enables efficient reuse of intermediate computations without compromising model performance. Extensive experiments on representative dLLMs, including LLaDA 8B and Dream 7B, show that dLLM-Cache achieves up to *9.1×* FLOPs reduction on LongBench-HotpotQA while maintaining competitive output quality. Notably, our method brings dLLM inference latency close to that of ARMs under many settings. *The code for this work is publicly available at:* `https://github.com/maomaocun/dLLM-cache`.

---

[*]Equal contribution [1]School of Artificial Intelligence, Shanghai Jiao Tong University, Shanghai, China [2]EPIC Lab, Shanghai Jiao Tong University, Shanghai, China [3]University of Electronic Science and Technology of China, Chengdu, China [4]Midea Group, China. Correspondence to: Linfeng Zhang <zhanglinfeng@sjtu.edu.cn>.

*Proceedings of the 43$^{rd}$ International Conference on Machine Learning*, Seoul, South Korea. PMLR 306, 2026. Copyright 2026 by the author(s).

## 1. Introduction

Large language models (LLMs) (Zhao et al., 2023) are foundational to modern AI, powering applications from conversational AI to scientific discovery. While autoregressive models (ARMs) have been the dominant paradigm (Radford et al., 2018; Brown et al., 2020; OpenAI, 2022), diffusion-based large language models (dLLMs), such as LLaDA (Nie et al., 2026) and Dream (Ye et al., 2025), have emerged as promising alternatives. These models offer impressive scalability and outperform ARMs in handling challenges like the "reversal curse" (Berglund et al., 2024) due to their bidirectional attention mechanism, demonstrating the potential of diffusion models for complex language tasks.

The practical adoption of dLLMs is hindered by a paradox: despite their potential for parallel decoding, they exhibit a daunting computational complexity of $\mathcal{O}(N^3)$. This inefficiency arises because generating a sequence of length $N$ requires $N$ denoising iterations in practice, each recalculating bidirectional attention across all tokens without any caching mechanism. This is fundamentally less efficient than standard ARMs, which exploit Key-Value caching (Pope et al., 2023) to reduce the overall computational effort to $\mathcal{O}(N^2)$.

Our work bridges this gap by successfully applying a caching mechanism to dLLMs. We first study two computational redundancies in the inference process of dLLMs as illustrated in Figure 1, which uniform strategies fail to address. First, **prompt redundancy** arises because the input prompt tokens remain constant, yet their internal representations, *e.g.*, attention output, are recomputed in each denoising step. Second, **response dynamics and redundancy** occur as the generated response features evolve. While significant similarity often exists between adjacent steps, suggesting caching potential, not all tokens evolve in the same way. This non-uniform evolution explains why traditional uniform caching strategies are ineffective.

Motivated by these insights, we introduce **dLLM-Cache**, a training-free, adaptive caching mechanism designed to accelerate dLLM inference by exploiting these distinct redundancies. dLLM-Cache employs a differentiated caching strategy comprising two core components:

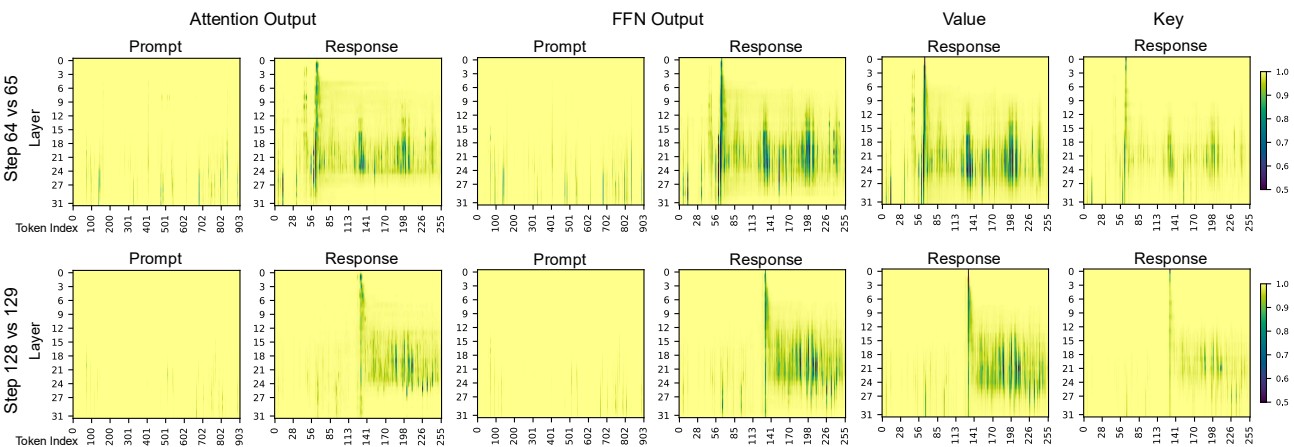

*Figure 1.* **Cosine similarity of Key, Value, Attention Output and FFN Output between two adjacent denoising steps in a dLLM**, highlighting computational redundancies. The heatmaps show similarity across adjacent steps for prompt and response tokens, respectively, where a lighter color indicates a higher similarity of a token compared with its value in the last step. These results demonstrate: (I) The prompt region exhibits high similarity, while the response region shows different similarity in different tokens. (II) Notably, only a small fraction of response tokens exhibit significantly lower similarity, suggesting that selective recomputation is sufficient. (III) Response tokens' value similarity closely aligns with attention and FFN output similarity, supporting that value changes can serve as an effective indicator to identify those most changed response tokens.

- **Long-Interval Prompt Caching:** We compute and cache features related to the prompt tokens only at sparse, long intervals, *e.g.*, every 100 steps. These cached features are then reused in all subsequent intermediate steps until the next long interval, drastically reducing the overhead associated with processing the static prompt.

- **Adaptive Short-Interval Response Caching:** Features associated with the response tokens are cached and fully refreshed at more frequent, shorter intervals, *e.g.*, every 10 steps. Between these full refreshes, we adopt an *adaptive partial update* strategy to balance speed and accuracy. Specifically, we identify and selectively update only the most dynamic tokens. As shown in Figure 1, the cosine similarity of a token's Value vector across adjacent steps strongly correlates with changes in its subsequent Attention and FFN Output. This motivates our **V-verify** mechanism, which uses Value similarity as an efficient proxy to select tokens for update.

This differentiated adaptive handling of prompt and response features allows significant inference acceleration while preserving quality, all without retraining. Our main contributions are:

1. We identify and characterize the dual computational redundancies in dLLM inference: quasi-static prompt and dynamic response redundancy.

2. We propose **dLLM-Cache**, a training-free adaptive caching framework that combines long-interval prompt caching with short-interval, similarity-guided partial updates for response tokens.

3. We introduce **V-verify**, a lightweight yet effective mechanism that uses cosine similarity of Value vectors across denoising steps to identify the most changed tokens for partial update, grounded in strong empirical correlation with overall token evolution.

4. We validate dLLM-Cache across various benchmarks, showing significant inference acceleration, *e.g.*, up to *9.1×* on LLaDA with **competitive output quality**, achieving a superior speed-quality trade-off compared to the baseline and simpler caching methods.

## 2. Related Work

### 2.1. Diffusion Models for Language.

Diffusion Models (DMs) (Sohl-Dickstein et al., 2015; Ho et al., 2020; Song et al., 2021) learn to reverse a data corruption process, excelling in continuous domains like images (Rombach et al., 2022; Peebles & Xie, 2023). However, adapting DMs to discrete data like text remains challenging, partly due to text's discrete nature. A promising direction involves Masked Diffusion Models (MDMs) (Austin et al., 2021; Lou et al., 2024; Shi et al., 2024; Ou et al., 2025; Zheng et al., 2024; Gong et al., 2025; Nie et al., 2025; He et al., 2023; Reid et al., 2023; Sahoo et al., 2024; Ye et al., 2023), a specific instance of discrete diffusion which operates on discrete sequences by iteratively predicting masked tokens based on their context.

Recent work has scaled MDMs (Nie et al., 2026; Ye et al., 2025), showing performance comparable to ARMs of similar size such as Llama 3 8B (Grattafiori et al., 2024). Their

bidirectional design helps mitigate limitations specific to ARMs like the reversal curse (Berglund et al., 2024), while extensions to multi-modal (Yang et al., 2026; You et al., 2025) and reasoning tasks (Zhao et al., 2026; Huang et al., 2026; Zhu et al., 2025) further highlight their versatility as a foundation model paradigm.

## 2.2. Acceleration via Caching Mechanisms

**Key–Value Caching in Autoregressive Models.** The most established use of caching in language models is the Key-Value (KV) caching (Pope et al., 2023), which is fundamental to the efficiency of ARMs. In ARMs, causal attention allows for the direct caching of past tokens' key and value states, trading memory for computational speed. However, cache size grows with input length, creating bottlenecks for long-context deployment. To address this, prior work sparsifies caches retrospectively (Xiao et al., 2024; Zhang et al., 2023; Ge et al., 2024; Liu et al., 2023; Li et al., 2024).

**Caching in Diffusion Language Models.** While feature caching has also been explored in ARMs, the bidirectional attention in dLLMs makes traditional KV caching incompatible (Nie et al., 2026), creating a distinct challenge. Prior works are beginning to address this gap, but often require cache-aware training (Arriola et al., 2025) or operate under restrictive conditions (Sahoo et al., 2024).

Concurrent works such as dKV-Cache (Ma et al., 2026) and Fast-dLLM (Wu et al., 2026) also study cache-enabled acceleration for dLLMs. The former focuses on delayed KV reuse after decoding, while the latter uses block-wise approximate KV Cache plus confidence-aware parallel decoding. However, dLLM-Cache exploits the asymmetric dynamics of prompt and response tokens during denoising, combining long-interval prompt caching with adaptive short-interval response caching guided by a lightweight V-verify mechanism. It also caches intermediate features beyond KV states, including Attention Output and FFN Output, to reduce repeated computations across Transformer blocks.

## 3. Methodology

### 3.1. Preliminary

**Training Paradigm of dLLMs.** Unlike the sequential and unidirectional nature of ARMs, dLLMs are trained in a denoising framework that learns to reverse a forward corruption process, where clean sequences are stochastically degraded over a continuous time variable.

Formally, let $\mathbf{x}_0 = (x_1, \ldots, x_L)$ be a clean text sequence sampled from the data distribution $\mathcal{D}$. The forward process defines a continuous time variable $t \in [0, 1]$, with $t = 0$ denoting the clean sequence and $t = 1$ the fully corrupted state. At each time $t$, a corrupted sequence $\mathbf{x}_t$ is produced,

where every token $x_{i,0}$ is independently transformed into $x_{i,t}$ according to the rule:

$$x_{i,t} = \begin{cases} [\text{MASK}] & \text{with probability } t \\ x_{i,0} & \text{with probability } 1 - t \end{cases} \quad (1)$$

This per-token independent masking process ensures that as $t \to 1$, the sequence $x_t$ converges to a fully masked state.

The model, a bidirectional Transformer parameterized by $\theta$ and denoted $p_\theta$, is trained to reconstruct the original sequence $x_0$ from its corrupted counterpart $x_t$. Training minimizes the negative log-likelihood of the original tokens at masked positions. Let $\mathcal{M}_t$ denote the indices of masked tokens in $x_t$. The loss is defined as:

$$\mathcal{L}(\theta) = -\mathbb{E}_{x_0 \sim \mathcal{D}, t \sim U[0,1]} \left[ \sum_{i \in \mathcal{M}_t} \log p_\theta(x_{i,0} | x_t) \right] \quad (2)$$

This training regimen compels the model to learn a robust representation of language structure by leveraging the full bidirectional context, rather than being constrained by a causal dependency chain.

**Inference Process of dLLMs.** dLLMs generate text via a non-autoregressive process that iteratively denoises a fully masked sequence into the target output. Our work focuses on accelerating this inference procedure. We use LLaDA as a representative example to illustrate it.

Let $\mathcal{T}$ be the token vocabulary and $[\text{MASK}] \in \mathcal{T}$ the special mask token. Given a prompt $\mathbf{c} = (c_1, \ldots, c_M)$, the model generates a response $\mathbf{y} = (y_1, \ldots, y_L)$ through $K$ discrete denoising steps, indexed by $k = K$ down to 0. Let $\mathbf{y}^{(k)} \in \mathcal{T}^L$ denote the intermediate state at step $k$, starting from a fully masked sequence:

$$\mathbf{y}^{(K)} = \big( \underbrace{[\text{MASK}], \ldots, [\text{MASK}]}_{L \text{ times}} \big) \quad (3)$$

At each step $k$, a mask predictor $p_\theta$ estimates the distribution over the clean sequence:

$$P_\theta(\mathbf{y} | \mathbf{c}, \mathbf{y}^{(k)}) = p_\theta(\mathbf{c}, \mathbf{y}^{(k)}; \theta) \quad (4)$$

The most likely clean sequence estimate $\hat{\mathbf{y}}^{(0|k)}$ is obtained via greedy decoding:

$$\hat{\mathbf{y}}^{(0|k)} = \arg\max_{\mathbf{y} \in \mathcal{T}^L} P_\theta(\mathbf{y} | \mathbf{c}, \mathbf{y}^{(k)}) \quad (5)$$

A transition function $S$ then yields $\mathbf{y}^{(k-1)}$ by selectively updating tokens in $\mathbf{y}^{(k)}$ based on $\hat{\mathbf{y}}^{(0|k)}$:

$$\mathbf{y}^{(k-1)} = S(\hat{\mathbf{y}}^{(0|k)}, \mathbf{y}^{(k)}, \mathbf{c}, k) \quad (6)$$

The specific strategy for $S$ may involve confidence-based remasking or semi-autoregressive block updates (Nie et al., 2026). While this process enables flexible generation, it incurs high latency due to repeated recomputation across steps, particularly as $K$ grows, as detailed in Appendix A.7.

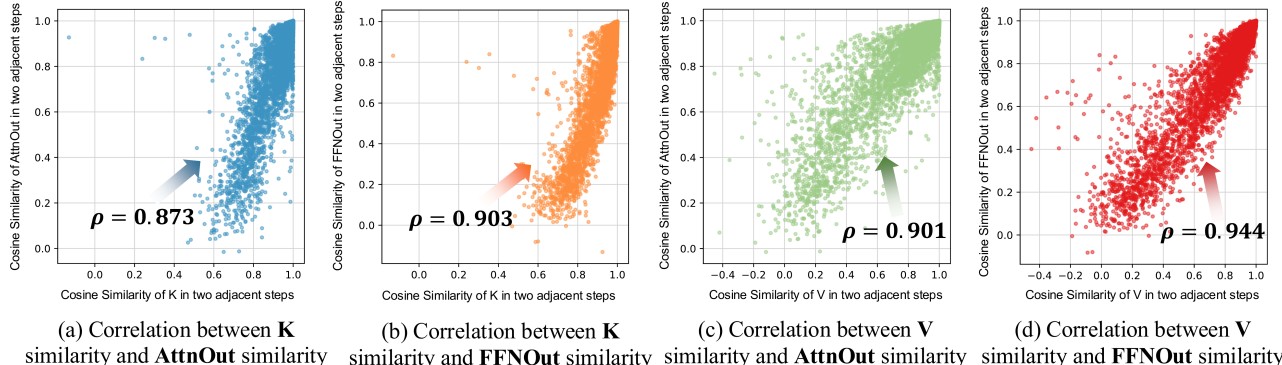

(a) Correlation between **K** similarity and **AttnOut** similarity

(b) Correlation between **K** similarity and **FFNOut** similarity

(c) Correlation between **V** similarity and **AttnOut** similarity

(d) Correlation between **V** similarity and **FFNOut** similarity

*Figure 2.* **Correlation of K/V and AttnOut/FFNOut similarities across adjacent denoising steps for response tokens.** We calculate the cosine similarity of the **K** or **V** vector for each response token between the current step and the previous cached step. This value is compared against the cosine similarity of the corresponding **AttnOut** or **FFNOut** across the same step pair. Taking the 25% most dissimilar tokens based on **K/V** similarity, we plot the correlation between these two metrics. The subfigures illustrate the relationships for **K** versus **AttnOut** (a), **K** versus **FFNOut** (b), **V** versus **AttnOut** (c), and **V** versus **FFNOut** (d).

### 3.2. dLLM-Cache

To alleviate the inference inefficiency of dLLMs, we introduce **dLLM-Cache**, a training-free caching framework. The input prompt remains static across denoising steps, and its internal features are consistently stable, making it suitable for long-interval caching. In contrast, the response sequence evolves dynamically. However, this evolution is highly sparse, as only a small fraction of response tokens change significantly at each step. Such sparsity, evident in Figure 1, suggests that recomputing all response features in every step is often unnecessary.

To take advantage of this sparsity, dLLM-Cache selectively updates only a small fraction of response tokens that change most between adjacent steps. The challenge is to identify such tokens efficiently and accurately. Figure 2 reveals that the change in a response token's Value (**V**) or Key (**K**) vector, which is quantified by cosine similarity between current and cached versions, strongly correlates with changes in its subsequent Attention Output (**AttnOut**) and Feedforward Network Output (**FFNOut**). This strong correlation indicates that by monitoring the dynamics of earlier-stage features like **V**, we can effectively identify tokens whose more complex downstream features are also likely to have significantly changed.

Based on this finding, we introduce our **V-verify** mechanism. It uses the cosine similarity between each response token's current **V** vector and its cached counterpart to identify tokens with the largest **V** changes. Only these selected tokens undergo a full feature recomputation and cache update.

Building on this core idea, the overall workflow of dLLM-Cache illustrated in Figure 3 is as follows: For each Transformer layer $l$, we store its $\mathbf{K}^{(l)}$, $\mathbf{V}^{(l)}$, $\mathbf{AttnOut}^{(l)}$, and $\mathbf{FFNOut}^{(l)}$ in a Prompt Cache $\mathcal{C}_p$ and a Response Cache $\mathcal{C}_r$, respectively. Caching is controlled by three hyperparam-

eters: prompt refresh interval $K_p$, response refresh interval $K_r$, and adaptive update ratio $\rho \in [0, 1]$. The inference process generally involves:

**Initialization.** At the very first step ($k = K$), we compute all features from $(\mathbf{c}, \mathbf{y}^{(K)})$, partitioning prompt-related features into $\mathcal{C}_p$ and response-related features into $\mathcal{C}_r$.

**Iterative Steps.** Next, as $k$ decreases from $K-1$ to 1, each layer $l$ performs the following operations:
(1) For the prompt, if $k \equiv 0 \pmod{K_p}$, recompute and update $\mathcal{C}_p$; otherwise, reuse.
(2) For the response, if $k \equiv 0 \pmod{K_r}$, fully recompute and update $\mathcal{C}_r$; otherwise, perform adaptive update detailed in Sec. 3.2.2.
(3) Each layer $l$ then continues the forward computation using the available feature version.

**Termination.** The process terminates at $k = 0$, yielding the final output $\mathbf{y}^{(0)}$.

#### 3.2.1. PROMPT CACHE MANAGEMENT

Since the input prompt $\mathbf{c}$ does not change, its features are largely stable over time. To take advantage of this, dLLM-Cache maintains a Prompt Cache $\mathcal{C}_p$. At $k = K$, all prompt-related features $\mathbf{K}_p^{(l)}, \mathbf{V}_p^{(l)}, \mathbf{AttnOut}_p^{(l)}, \mathbf{FFNOut}_p^{(l)}$ are computed and stored. In subsequent steps, these features are recomputed only every $K_p$ steps; in other steps, they are reused directly from the cache. This reduces the cost of processing the static prompt, particularly when $K_p$ is large.

#### 3.2.2. RESPONSE CACHE WITH ADAPTIVE UPDATES

Response features $\mathbf{y}^{(k)}$ evolve over time, though most tokens change only gradually across adjacent steps, allowing selective updates. The response cache $\mathcal{C}_r$ supports two modes: periodic full refresh and adaptive partial update.

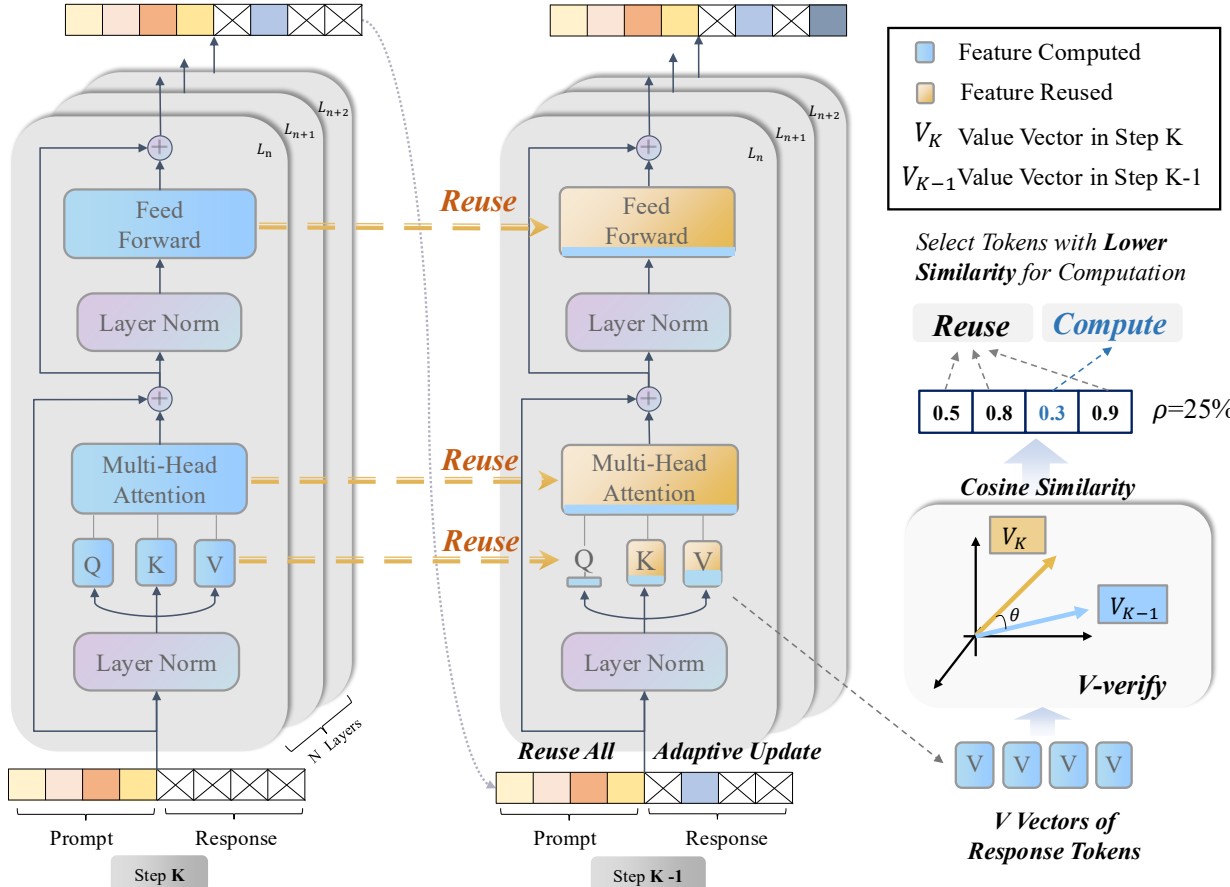

*Figure 3.* **The dLLM-Cache pipeline.** Prompt features are updated at long intervals, while response features are fully recomputed every $K_r$ steps and otherwise only the tokens least similar to cached $\mathbf{V}$ are recomputed.

**Full Refresh.** All response features are recomputed when $k \equiv 0 \pmod{K_r}$ or $k = K$.

**Adaptive Partial Update.** Otherwise, we first compute the cosine similarity $s_j$ between the current Value vector $\mathbf{v}_{r,j}^{(l)}$ and its cached counterpart $\tilde{\mathbf{v}}_{r,j}^{(l)}$ for each token $j$ (Eq. 7).

$$s_j = \frac{(\mathbf{v}_{r,j}^{(l)})^\top \tilde{\mathbf{v}}_{r,j}^{(l)}}{\|\mathbf{v}_{r,j}^{(l)}\| \|\tilde{\mathbf{v}}_{r,j}^{(l)}\|} \qquad (7)$$

Then we select the $\lfloor \rho |\mathbf{y}^{(k)}| \rfloor$ tokens with the lowest similarity for updating, recompute their features, and reuse cached values for the rest. In adaptive partial updates, V-verify computes current Value vectors for all response tokens to score adjacent-step changes. Since this full response-side $\mathbf{V}_r^{\text{new}}$ is already available after the lightweight projection, we overwrite the entire cached $\mathbf{V}_r$ with $\mathbf{V}_r^{\text{new}}$. In contrast, $\mathbf{K}$, **AttnOut**, and **FFNOut** are recomputed and scattered back only for the selected low-similarity tokens.

This adaptive strategy leverages temporal stability to cut computation while preserving accuracy.

## 4. Experiments

### 4.1. Experiment Settings

**Implementation Details.** We evaluated dLLM-Cache on two representative dLLMs: LLaDA 8B (Nie et al., 2026) and Dream 7B (Ye et al., 2025), each with Base and Instruct variants. Following the original inference configurations detailed in Appendix A.11, we conducted our experiments across eight benchmarks. Additional performance analysis on long and semantically diverse scenarios is provided in Appendix A.1. For all models, we set $\rho = 0.25$. The $K_p$ and $K_r$ are specified in Appendix A.11. All experiments were conducted on the NVIDIA RTX 4090 GPUs.

**Evaluation Metrics.** We evaluate the acceleration and model quality preservation of dLLM-Cache using several metrics. Throughput is measured as Tokens Per Second (TPS), reflecting inference speed. Computational cost is calculated as the average Floating Point Operations (FLOPs) per token. Task performance is assessed using benchmark scores, like accuracy on GSM8K (Cobbe et al., 2021), ensuring dLLM-Cache achieves efficiency gains without compromising model performance. The testing of TPS and FLOPs was performed on a single RTX 4090 GPU.

*Table 1.* **Comparison of LLaDA 8B with and without dLLM-Cache** on 8 benchmarks.

| Task | Method | Inference Efficiency | | | | Performance |
|---|---|---|---|---|---|---|
| | | TPS↑ | Speed(TPS)↑ | FLOPs(T)↓ | Speed(FLOPs)↑ | Score↑ |
| **Mathematics & Science** | | | | | | |
| GSM8K | LLaDA Base | 7.32 | 1.00× | 16.12 | 1.00× | 69.06 |
| | + dLLM-Cache | 23.19$_{+15.87}$ | 3.17×$_{+2.17}$ | 3.21$_{-12.91}$ | 5.02×$_{+4.02}$ | 70.66$_{+1.60}$ |
| | LLaDA Instruct | 6.95 | 1.00× | 16.97 | 1.00× | 77.48 |
| | + dLLM-Cache | 29.75$_{+22.80}$ | 4.28×$_{+3.28}$ | 2.92$_{-14.05}$ | 5.81×$_{+4.81}$ | 78.54$_{+1.06}$ |
| GPQA | LLaDA Base | 5.12 | 1.00× | 22.97 | 1.00× | 31.91 |
| | + dLLM-Cache | 25.23$_{+20.11}$ | 4.93×$_{+3.93}$ | 3.20$_{-19.77}$ | 7.18×$_{+6.18}$ | 32.81$_{+0.90}$ |
| | LLaDA Instruct | 5.33 | 1.00× | 22.07 | 1.00× | 29.01 |
| | + dLLM-Cache | 28.01$_{+22.68}$ | 5.26×$_{+4.26}$ | 2.73$_{-19.34}$ | 8.08×$_{+7.08}$ | 29.01$_{+0.00}$ |
| Math | LLaDA Base | 8.31 | 1.00× | 14.11 | 1.00× | 30.84 |
| | + dLLM-Cache | 33.92$_{+25.61}$ | 4.08×$_{+3.08}$ | 2.61$_{-11.50}$ | 5.41×$_{+4.41}$ | 29.84$_{-1.00}$ |
| | LLaDA Instruct | 23.65 | 1.00× | 5.16 | 1.00× | 22.32 |
| | + dLLM-Cache | 31.02$_{+7.37}$ | 1.31×$_{+0.31}$ | 3.96$_{-1.20}$ | 1.30×$_{+0.30}$ | 22.52$_{+0.20}$ |
| **General Tasks** | | | | | | |
| MMLU-pro | LLaDA Base | 14.08 | 1.00× | 8.40 | 1.00× | 24.26 |
| | + dLLM-Cache | 45.75$_{+31.67}$ | 3.25×$_{+2.25}$ | 2.15$_{-6.25}$ | 3.91×$_{+2.91}$ | 24.69$_{+0.43}$ |
| | LLaDA Instruct | 14.01 | 1.00× | 8.46 | 1.00× | 36.41 |
| | + dLLM-Cache | 39.63$_{+25.62}$ | 2.83×$_{+1.83}$ | 2.62$_{-5.84}$ | 3.23×$_{+2.23}$ | 36.08$_{-0.33}$ |
| MMLU | LLaDA Base | 8.09 | 1.00× | 14.56 | 1.00× | 63.99 |
| | + dLLM-Cache | 33.52$_{+25.43}$ | 4.14×$_{+3.14}$ | 2.64$_{-11.92}$ | 5.52×$_{+4.52}$ | 64.26$_{+0.27}$ |
| | LLaDA Instruct | 10.12 | 1.00× | 11.85 | 1.00× | 61.24 |
| | + dLLM-Cache | 21.23$_{+11.11}$ | 2.10×$_{+1.10}$ | 4.50$_{-7.35}$ | 2.63×$_{+1.63}$ | 62.82$_{+1.58}$ |
| BBH | LLaDA Base | 6.41 | 1.00× | 18.29 | 1.00× | 44.77 |
| | + dLLM-Cache | 27.90$_{+21.49}$ | 4.35×$_{+3.35}$ | 3.09$_{-15.20}$ | 5.92×$_{+4.92}$ | 45.04$_{+0.27}$ |
| | LLaDA Instruct | 6.18 | 1.00× | 18.98 | 1.00× | 51.49 |
| | + dLLM-Cache | 27.55$_{+21.37}$ | 4.46×$_{+3.46}$ | 3.08$_{-15.90}$ | 6.16×$_{+5.16}$ | 51.98$_{+0.49}$ |
| **Code** | | | | | | |
| MBPP | LLaDA Base | 7.87 | 1.00× | 14.91 | 1.00× | 40.80 |
| | + dLLM-Cache | 24.61$_{+16.74}$ | 3.13×$_{+2.13}$ | 3.07$_{-11.84}$ | 4.86×$_{+3.86}$ | 40.60$_{-0.20}$ |
| | LLaDA Instruct | 7.55 | 1.00× | 15.53 | 1.00× | 39.20 |
| | + dLLM-Cache | 31.73$_{+24.18}$ | 4.20×$_{+3.20}$ | 2.80$_{-12.73}$ | 5.55×$_{+4.55}$ | 39.60$_{+0.40}$ |
| HumanEval | LLaDA Base | 19.98 | 1.00× | 6.03 | 1.00× | 32.92 |
| | + dLLM-Cache | 51.96$_{+31.98}$ | 2.60×$_{+1.60}$ | 2.04$_{-3.99}$ | 2.96×$_{+1.96}$ | 32.31$_{-0.61}$ |
| | LLaDA Instruct | 10.57 | 1.00× | 11.10 | 1.00× | 38.71 |
| | + dLLM-Cache | 44.77$_{+34.20}$ | 4.24×$_{+3.24}$ | 2.05$_{-9.05}$ | 5.41×$_{+4.41}$ | 39.02$_{+0.31}$ |

## 4.2. Main Results

**Performance and Efficiency Gains across Models.** Tables 1 and 2 summarize the results for LLaDA 8B and Dream 7B. Across tasks, dLLM-Cache consistently improves inference efficiency while largely preserving task performance. For LLaDA 8B, applying dLLM-Cache to LLaDA Instruct on GPQA yields an 8.08× FLOPs speedup, reducing the cost from 22.07T to 2.73T without accuracy degradation. The same trend holds for Dream 7B. These results demonstrate that dLLM-Cache generalizes across model families and benchmark categories, providing broad acceleration on multiple tasks.

**Comparison with Other Representative LLM.** Table 3 compares our accelerated LLaDA with Llama 3 8B (Grattafiori et al., 2024). Note this is merely an efficiency reference given differing training data. Simply reducing denoising steps to 32 boosts throughput but catastrophically drops accuracy to 22.25%. In contrast, dLLM-Cache achieves a 2.8× speedup while maintaining 70.66% accuracy. When further combined with SlowFast Sampling (Wei et al., 2026), an advanced sampling method, throughput improves to 49.86 TPS, comparable to Llama 3 8B while re-

*Table 2.* **Comparison of Dream 7B with and without dLLM-Cache** on 8 benchmarks.

| Task | Configuration | Inference Efficiency | | | | Performance |
|------|---------------|------|------|------|------|------|
| | | TPS↑ | Speed(TPS)↑ | FLOPs(T)↓ | Speed(FLOPs)↑ | Score↑ |
| **Mathematics & Science** | | | | | | |
| GSM8K | Dream Base | 6.36 | 1.00× | 19.59 | 1.00× | 76.95 |
| | + dLLM-Cache | 32.44 $_{+26.08}$ | 5.10× $_{+4.10}$ | 2.84 $_{-16.75}$ | 6.90× $_{+5.90}$ | 76.95 $_{+0.00}$ |
| | Dream Instruct | 6.39 | 1.00× | 19.57 | 1.00× | 77.55 |
| | + dLLM-Cache | 24.52 $_{+18.13}$ | 3.84× $_{+2.84}$ | 4.24 $_{-15.33}$ | 4.62× $_{+3.61}$ | 76.80 $_{-0.75}$ |
| GPQA | Dream Base | 5.80 | 1.00× | 21.66 | 1.00× | 33.92 |
| | + dLLM-Cache | 30.95 $_{+25.15}$ | 5.33× $_{+4.33}$ | 3.03 $_{-18.63}$ | 7.15× $_{+6.15}$ | 34.15 $_{+0.23}$ |
| | Dream Instruct | 5.53 | 1.00× | 22.63 | 1.00× | 34.38 |
| | + dLLM-Cache | 21.98 $_{+16.45}$ | 3.97× $_{+2.97}$ | 4.69 $_{-17.94}$ | 4.83× $_{+3.82}$ | 33.93 $_{-0.45}$ |
| Math | Dream Base | 9.40 | 1.00× | 13.31 | 1.00× | 38.68 |
| | + dLLM-Cache | 36.89 $_{+27.49}$ | 3.92× $_{+2.92}$ | 2.61 $_{-10.70}$ | 5.10× $_{+4.10}$ | 37.94 $_{-0.74}$ |
| | Dream Instruct | 8.85 | 1.00× | 14.11 | 1.00× | 38.28 |
| | + dLLM-Cache | 23.52 $_{+14.67}$ | 2.66× $_{+1.66}$ | 4.66 $_{-9.45}$ | 3.03× $_{+2.03}$ | 37.62 $_{-0.66}$ |
| **General Tasks** | | | | | | |
| MMLU-pro | Dream Base | 15.61 | 1.00× | 7.92 | 1.00× | 24.13 |
| | + dLLM-Cache | 35.86 $_{+20.25}$ | 2.30× $_{+1.30}$ | 2.89 $_{-5.03}$ | 2.74× $_{+1.74}$ | 23.86 $_{-0.27}$ |
| | Dream Instruct | 15.40 | 1.00× | 7.98 | 1.00× | 43.79 |
| | + dLLM-Cache | 23.98 $_{+8.58}$ | 1.56× $_{+0.56}$ | 4.77 $_{-3.21}$ | 1.67× $_{+0.67}$ | 43.96 $_{+0.17}$ |
| MMLU | Dream Base | 9.10 | 1.00× | 13.73 | 1.00× | 73.49 |
| | + dLLM-Cache | 31.07 $_{+21.97}$ | 3.41× $_{+2.41}$ | 3.27 $_{-10.46}$ | 4.20× $_{+3.20}$ | 73.20 $_{-0.29}$ |
| | Dream Instruct | 8.45 | 1.00× | 14.75 | 1.00× | 73.40 |
| | + dLLM-Cache | 38.01 $_{+29.56}$ | 4.50× $_{+3.50}$ | 2.42 $_{-12.33}$ | 6.10× $_{+5.10}$ | 73.42 $_{+0.02}$ |
| BBH | Dream Base | 7.24 | 1.00× | 17.25 | 1.00× | 52.25 |
| | + dLLM-Cache | 29.61 $_{+22.37}$ | 4.09× $_{+3.09}$ | 3.35 $_{-13.90}$ | 5.15× $_{+4.15}$ | 51.66 $_{-0.59}$ |
| | Dream Instruct | 6.98 | 1.00× | 17.90 | 1.00× | 57.07 |
| | + dLLM-Cache | 22.31 $_{+15.33}$ | 3.20× $_{+2.20}$ | 4.82 $_{-13.08}$ | 3.71× $_{+2.71}$ | 57.07 $_{+0.00}$ |
| **Code** | | | | | | |
| MBPP | Dream Base | 8.91 | 1.00× | 14.06 | 1.00× | 54.20 |
| | + dLLM-Cache | 35.69 $_{+26.78}$ | 4.01× $_{+3.01}$ | 2.66 $_{-11.40}$ | 5.29× $_{+4.29}$ | 54.20 $_{+0.00}$ |
| | Dream Instruct | 8.46 | 1.00× | 14.65 | 1.00× | 57.00 |
| | + dLLM-Cache | 29.77 $_{+21.31}$ | 3.52× $_{+2.52}$ | 3.33 $_{-11.32}$ | 4.40× $_{+3.40}$ | 56.80 $_{-0.20}$ |
| HumanEval | Dream Base | 21.43 | 1.00× | 5.68 | 1.00× | 58.53 |
| | + dLLM-Cache | 27.40 $_{+5.97}$ | 1.28× $_{+0.28}$ | 4.17 $_{-1.51}$ | 1.36× $_{+0.36}$ | 57.31 $_{-1.22}$ |
| | Dream Instruct | 17.88 | 1.00× | 6.84 | 1.00× | 57.92 |
| | + dLLM-Cache | 28.03 $_{+10.15}$ | 1.57× $_{+0.57}$ | 3.94 $_{-2.90}$ | 1.74× $_{+0.74}$ | 56.09 $_{-1.83}$ |

taining 67.17% accuracy, surpassing it by 18.12%, showing the orthogonality of our method.

*Table 3.* **Comparison of LLaDA 8B Base with other representative LLM** on GSM8K.

| Method | Steps | TPS↑ | Acc (%)↑ | Memory (GB)↓ |
|--------|-------|------|----------|--------------|
| Llama 3 8B | - | 47.73 | 49.05 | 16.06 |
| LLaDA Base | 32 | 53.55 | 22.25 | 16.94 |
| LLaDA Base | 256 | 7.37 | 69.06 | 16.94 |
| + Cache | 256 | 20.64 | **70.66** | 17.93 |
| + Cache + SlowFast | - | 49.86 | 67.17 | 17.93 |

**Compatibility with Advanced Sampling Methods.** Our dLLM-Cache is orthogonal to recent sampling-based accel-

eration methods, such as SlowFast Sampling (Wei et al., 2026). When combined, as shown in Table 4, the two methods achieve greater inference speedups while preserving model performance.

**Comparison with Concurrent Caching Methods.** We compared dLLM-Cache with two **concurrent** works, dKV-Cache (Ma et al., 2026) and Fast-dLLM (Wu et al., 2026), as detailed in Table 5. While these approaches also aim to accelerate inference, they primarily rely on coarse-grained token heuristics and limit their scope to caching KV pairs, often leaving the computationally intensive FFNs unoptimized. In contrast, dLLM-Cache leverages the fine-grained

*Table 4.* **Performance of LLaDA Base with dLLM-Cache and SlowFast Sampling.**

| Task | Method | TPS↑ | Speed↑ | Score↑ |
|------|--------|------|--------|--------|
| **Mathematics & Science** | | | | |
| GSM8K | LLaDA Base | 4.55 | 1.00× | 69.83 |
| | Sampling + Cache | **26.99** | **5.93×** | **69.60** |
| GPQA | LLaDA Base | 3.31 | 1.00× | 31.47 |
| | Sampling + Cache | **29.06** | **8.78×** | **33.48** |
| **General Tasks** | | | | |
| MMLU-pro | LLaDA Base | 9.16 | 1.00× | 23.30 |
| | Sampling + Cache | **33.38** | **3.64×** | **25.53** |
| BBH | LLaDA Base | 4.04 | 1.00× | 44.97 |
| | Sampling + Cache | **36.04** | **8.92×** | **44.81** |
| **Code** | | | | |
| HumanEval | LLaDA Base | 11.24 | 1.00× | 31.71 |
| | Sampling + Cache | **41.14** | **3.66×** | **31.10** |

*Table 5.* **Comparison of dLLM-Cache with other *concurrent* methods on LLaDA and Dream.**

| Task | Method | TPS↑ | Speed↑ | Score↑ |
|------|--------|------|--------|--------|
| GPQA | Dream Base | 5.80 | 1.00× | 33.92 |
| | + dKV-Cache | 10.11 | 1.74× | 32.83 |
| | + Fast-dLLM | 22.23 | 3.83× | 31.31 |
| | + dLLM-Cache | **30.95** | **5.33×** | **34.15** |
| MMLU | LLaDA Instruct | 10.12 | 1.00× | 61.24 |
| | + dKV-Cache | 14.34 | 1.42× | 60.87 |
| | + Fast-dLLM | 20.51 | 2.03× | 61.43 |
| | + dLLM-Cache | **21.23** | **2.10×** | **62.82** |
| HumanEval | LLaDA Instruct | 10.57 | 1.00× | 38.71 |
| | + dKV-Cache | 14.40 | 1.36× | 37.20 |
| | + Fast-dLLM | 21.50 | 2.03× | 36.59 |
| | + dLLM-Cache | **44.77** | **4.24×** | **39.02** |

**V-verify** mechanism to identify feature stability, enabling the safe bypassing of **entire transformer layers**, including both Attention and FFN projections. This advantage translates directly into superior efficiency: on GPQA with Dream Base, dLLM-Cache achieves a **5.33× speedup**, significantly outperforming the 1.74× and 3.83× gains of the baselines while maintaining **the highest accuracy** of 34.15%. A more detailed comparison is provided in Appendix A.2.

### 4.3. Ablation Study

**Effect of Update Ratio $\rho$ and Selection Strategy.** We investigated how different token selection strategies impact performance under varying adaptive update ratios $\rho$. Figure 4 reports accuracy and computational cost on GSM8K when using three strategies: **V-verify**, **K-verify**, and ran-

dom selection. Both similarity-based strategies consistently outperform random selection across a wide range of $\rho$ values, confirming the importance of dynamic, feature-driven updates. In particular, value-based selection achieves the highest accuracy around $\rho = 0.25$, while requiring significantly fewer FLOPs than full recomputation. This suggests that moderate, targeted updates, *e.g.*, $\rho \approx 0.25$ strike a favorable **trade-off** between efficiency and output quality.

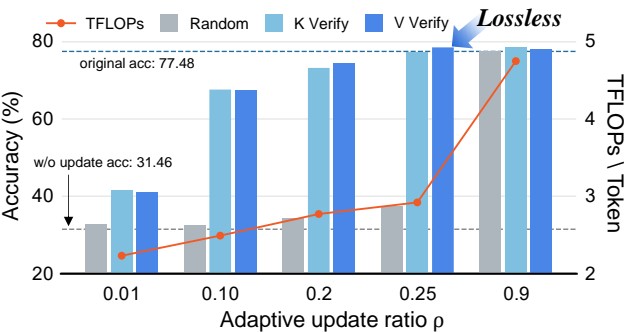

*Figure 4.* **Effect of token selection strategy** on GSM8K using LLaDA 8B Instruct model under varying update ratios $\rho$.

**Effect of Cache Refresh Interval $K_p$ and $K_r$.** We analyzed how refresh intervals affect efficiency and accuracy. As shown in Figure 5(a), increasing the prompt interval $K_p$ substantially reduces FLOPs without hurting accuracy, confirming that infrequent prompt updates suffice. Figure 5(b) highlights the importance of response updates. Without prompt caching or adaptive updates ($K_p = 1$, $\rho = 0$, gray line), accuracy drops sharply. In contrast, our setting ($K_p = 50$, $\rho = 0.25$, orange and blue line) maintains high accuracy with much lower cost. This validates our strategy of combining long prompt intervals with short response intervals and adaptive updates. Additional analyses of the Dream model can be found in Appendix A.3.

## 5. Discussion

**Effect of Denoising Steps.** In dLLMs, the number of denoising steps determines a trade-off between quality and latency. Increasing the steps improves output accuracy but also raises inference cost, as shown in Figure 5(c). Simply reducing the steps accelerates inference but causes severe performance degradation. On GSM8K, dLLM-Cache achieves a 5× lossless speedup at 256 steps, matching the computational cost of a baseline with approximately 48 steps while more than doubling its accuracy. This shows that our method achieves both efficiency and quality, unlike simple step reduction.

**Storage Overhead of Caching.** dLLM-Cache stores four types of intermediate features per layer: **K**, **V**, **AttnOut**, and **FFNOut**. The total cache size scales with the number of tokens $T$, embedding dimension $d$, and number of layers $L$, giving a cost of $T \times d \times 4 \times L$ as detailed in Appendix A.9.

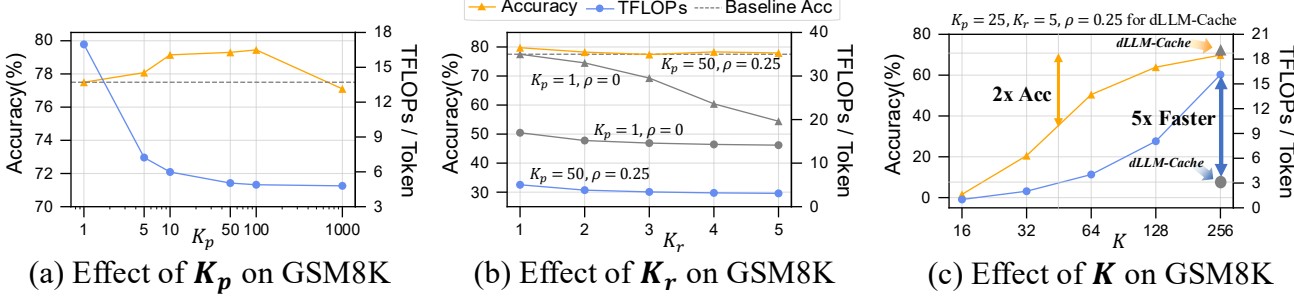

(a) Effect of $K_p$ on GSM8K  (b) Effect of $K_r$ on GSM8K  (c) Effect of $K$ on GSM8K

*Figure 5.* (a) Varying $K_p$ with $K_r = 1$, $\rho = 0$. (b) Varying $K_r$ under two settings: baseline ($K_p = 1$, $\rho = 0$, gray) and our setup ($K_p = 50$, $\rho = 0.25$ in Table 1). (c) Varying the denoising steps $K$ for the uncached baseline. The gray marker denotes dLLM-Cache with $K_p = 25$, $K_r = 5$, $\rho = 0.25$. dLLM-Cache achieves a better Pareto optimum: about $5\times$ faster than the standard 256-step baseline at similar accuracy, and over $2\times$ more accurate than a baseline with equal FLOPs. (a–b) LLaDA Instruct; (c) LLaDA Base.

Since only one version per layer is cached, the overall footprint remains stable. As shown in Table 3, on GSM8K with LLaDA 8B Base, peak GPU usage is 16.94 GB without caching, 17.93 GB with dLLM-Cache, and 16.06 GB for Llama 3 8B. Hence, the additional memory cost introduced by caching is less than 1 GB, representing only a 5% increase over the baseline. This modest storage overhead yields up to $9\times$ acceleration in generation, making it a highly favorable tradeoff, particularly under tight latency constraints and limited GPU memory budgets.

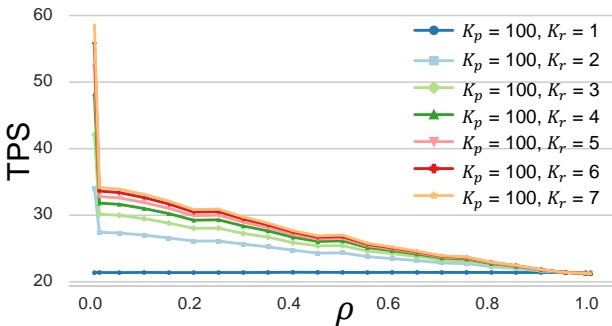

*Figure 6.* **TPS versus $\rho$.** A notable decrease in TPS at minimal $\rho$ reflects the fixed cost of initiating selective updates.

**Cost of V-verify and the Fixed Update Overheads.** Our **V-verify** mechanism uses lightweight **V** vector similarity for identifying dynamic tokens. While **V-verify** itself is computationally inexpensive, practical speedup from adaptive partial updates is constrained by fixed operational overheads. Figure 6 shows that there is a latency penalty when transitioning from a fully cached state to adaptive updates ($\rho > 0$). This base cost arises because initiating any selective recomputation triggers non-negligible system-level latencies, *e.g.*, for GPU kernel management and data movement that are not strictly proportional to the number of updated tokens. Consequently, at very low $\rho$ values, these fixed overheads dominate, limiting further run time savings. An optimal $\rho$ must balance these fixed costs against saved dynamic computation, while preserving model quality. Figure 4 suggests

$\rho \approx 0.25$ offers an effective trade-off between the costs of activating selective updates and the benefits of reduced computation, optimizing overall efficiency and fidelity.

## 6. Conclusion

We present dLLM-Cache, a training-free and model-agnostic caching method for accelerating inference in diffusion-based large language models. Extensive evaluations on LLaDA and Dream confirm that dLLM-Cache delivers up to *9.1×* FLOPs reduction while maintaining competitive generation quality.

## Impact Statement

This paper presents work whose goal is to advance the field of machine learning. There are many potential societal consequences of our work, none of which we feel must be specifically highlighted here.

## Acknowledgement

This work was supported by the CCF-Tencent Rhino-Bird Funds.

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

# A. Appendix

## A.1. Performance on Long and Semantically Diverse Scenarios

**Efficiency on Extensive Inputs.** Our *Long-Interval Prompt Caching* mechanism is particularly advantageous for scenarios involving long static prompts. For example, on the LongBench-HotpotQA (Bai et al., 2024) task using the LLaDA 8B Base model, dLLM-Cache achieves a **9.1×** **FLOPs reduction** over the baseline. Remarkably, this acceleration is accompanied by a performance gain, with the F1 score increasing from 34.56 to 36.10, demonstrating that our method can maximally leverage prompt stability without sacrificing precision.

*Table 6.* **Comparison of LongBench performance on LLaDA Instruct and Dream Instruct with and without dLLM-Cache.**

| Method | Single-Doc. QA | | Multi-Doc. QA | | | Summarization | | | Few-shot Learning | | | Synthetic | Code | | Ave. Score |
| --- | --- | --- | --- | --- | --- | --- | --- | --- | --- | --- | --- | --- | --- | --- | --- |
| | Qasper | MF-en | HotpotQA | 2WikiMQA | Musique | GovReport | QMSum | MultiNews | TREC | TriviaQA | SAMSum | PRe | Lcc | RB-P | |
| LLaDA Instruct | 16.96 | 31.31 | 14.68 | 17.60 | 11.48 | 29.24 | 21.93 | 27.58 | 65.20 | 47.98 | 40.51 | 98.17 | 65.69 | 59.57 | **39.14** |
| + dLLM-Cache | 15.26 | 29.62 | 13.87 | 17.17 | 10.44 | 29.75 | 22.06 | 26.68 | 66.00 | 44.94 | 41.86 | 97.44 | 66.07 | 59.34 | **38.61** |
| Dream Instruct | 28.17 | 36.23 | 27.65 | 32.43 | 11.83 | 5.04 | 14.29 | 5.95 | 73.00 | 89.25 | 37.84 | 16.92 | 38.91 | 45.08 | **33.04** |
| + dLLM-Cache | 26.55 | 39.86 | 27.66 | 32.09 | 11.12 | 4.40 | 13.89 | 5.51 | 73.50 | 89.59 | 36.07 | 12.05 | 39.88 | 45.57 | **32.70** |

**Robustness Across Diverse Tasks.** To verify that this efficiency does not compromise general capabilities in complex real-world scenarios, we evaluated dLLM-Cache on the comprehensive LongBench benchmark using Instruct models. Table 6 details the performance across six major task categories. dLLM-Cache demonstrates strong performance retention where the LLaDA Instruct model maintains an average score of 38.61 compared to the 39.14 baseline. Similarly, Dream Instruct achieves a score of 32.70 against the baseline of 33.04. These results confirm that dLLM-Cache effectively preserves deep semantic understanding and long-range dependency reasoning capabilities across a wide range of tasks.

## A.2. Detailed Comparison with Concurrent Caching Methods

While Table 5 presents a compact overview, we provide here a more comprehensive comparison between dLLM-Cache and the concurrent caching methods dKV-Cache (Ma et al., 2026) and Fast-dLLM (Wu et al., 2026) in Table 7.

*Table 7.* **Comparison of dLLM-Cache with other *concurrent* methods on LLaDA (left) and Dream (right).**

| Task | Method | TPS↑ | Speed↑ | Memory↓ | Score↑ | Task | Method | TPS↑ | Speed↑ | Memory↓ | Score↑ |
| --- | --- | --- | --- | --- | --- | --- | --- | --- | --- | --- | --- |
| GSM8K | LLaDA Instruct | 6.95 | 1.00× | 15.86 | 77.48 | GSM8K | Dream Base | 6.36 | 1.00× | 15.73 | 76.95 |
| | + dKV-Cache | 8.89 | 1.28× | 21.08 | **79.30** | | + dKV-Cache | 10.26 | 1.61× | **16.14** | 76.57 |
| | + Fast-dLLM | 19.11 | 2.75× | 19.48 | 75.89 | | + Fast-dLLM | 21.36 | 2.08× | 19.95 | 74.30 |
| | + dLLM-Cache | **29.75** | **4.28×** | **17.85** | 78.54 | | + dLLM-Cache | **32.44** | **5.10×** | 16.76 | **76.95** |
| MMLU | LLaDA Instruct | 10.12 | 1.00× | 15.54 | 61.24 | GPQA | Dream Base | 5.80 | 1.00× | 15.77 | 33.92 |
| | + dKV-Cache | 14.34 | 1.42× | 17.88 | 60.87 | | + dKV-Cache | 10.11 | 1.74× | **16.23** | 32.83 |
| | + Fast-dLLM | 20.51 | 2.03× | 17.13 | 61.43 | | + Fast-dLLM | 22.23 | 3.83× | 20.69 | 31.31 |
| | + dLLM-Cache | **21.23** | **2.10×** | **16.61** | **62.82** | | + dLLM-Cache | **30.95** | **5.33×** | 16.93 | **34.15** |
| HumanEval | LLaDA Instruct | 10.57 | 1.00× | 15.39 | 38.71 | MMLU | Dream Base | 9.10 | 1.00× | 15.64 | 73.49 |
| | + dKV-Cache | 14.40 | 1.36× | 17.17 | 37.20 | | + dKV-Cache | 12.80 | 1.41× | **15.92** | 72.77 |
| | + Fast-dLLM | 21.50 | 2.03× | **16.60** | 36.59 | | + Fast-dLLM | 23.69 | 2.60× | 18.32 | 72.69 |
| | + dLLM-Cache | **44.77** | **4.24×** | 16.65 | **39.02** | | + dLLM-Cache | **31.07** | **3.41×** | 16.37 | **73.20** |

## A.3. Detailed Sensitivity Analysis on Dream 7B

As demonstrated in the main paper, dLLM-Cache is effective across different dLLM architectures, including both LLaDA and Dream. This highlights the generalizability of our approach, which targets computational redundancies fundamental to the diffusion process rather than model-specific artifacts.

To further substantiate the robustness of our method and provide deeper insight into its behavior, this section presents a detailed sensitivity analysis of dLLM-Cache's key hyperparameters when applied to the Dream 7B model. The results,

shown in Table 8, Table 9, and Table 10, reveal performance trends that are highly consistent with those observed for LLaDA. This confirms the stable and predictable behavior of our method across different models.

*Table 8.* Sensitivity analysis of the adaptive update ratio $\rho$ on Dream 7B for the GPQA benchmark. Hyperparameters are set to $K_p = 25$ and $K_r = 4$.

| $\rho$ | 0 | 0.1 | 0.2 | 0.25 | 0.3 | 0.5 | 0.75 | 1 |
|---|---|---|---|---|---|---|---|---|
| Accuracy (%) | 35.04 | 36.16 | 35.93 | 35.04 | 35.04 | 34.59 | 35.49 | 35.26 |

*Table 9.* Sensitivity analysis of the prompt refresh interval $K_p$ on Dream 7B for the GPQA benchmark. Hyperparameters are set to $K_r = 4$ and $\rho = 0.25$.

| $K_p$ | 10 | 25 | 50 | 100 |
|---|---|---|---|---|
| Accuracy (%) | 35.04 | 35.04 | 35.04 | 35.04 |

*Table 10.* Sensitivity analysis of the response refresh interval $K_r$ on Dream 7B for the GPQA benchmark. Hyperparameters are set to $K_p = 25$ and $\rho = 0.25$.

| $K_r$ | 2 | 4 | 6 |
|---|---|---|---|
| Accuracy (%) | 36.16 | 35.04 | 33.92 |

### A.4. Cross-Layer Behavior of Token Selection

The layer-wise token selection in dLLM-Cache is a natural consequence of its cache design. Cached features are stored separately for each Transformer layer $l$. At each layer, we cache $\mathbf{K}^{(l)}$, $\mathbf{V}^{(l)}$, $\mathbf{AttnOut}^{(l)}$, and $\mathbf{FFNOut}^{(l)}$ in the Prompt Cache $\mathcal{C}_p$ and the Response Cache $\mathcal{C}_r$. The V-verify score is therefore defined locally at layer $l$ by comparing the current Value vector with its cached version, which makes per-layer token selection the most consistent choice for the cache design.

The selected token set is expected to remain largely stable across adjacent layers. Let $s_j^{(l)}$ denote the V-verify score of response token $j$ at layer $l$. Under standard smoothness assumptions, if the layer map and the Value projection are Lipschitz, then $|s_j^{(l)} - s_j^{(l+1)}| \leq \varepsilon$ for some small $\varepsilon$. Let $\Delta$ denote the margin at the selection boundary, where $\Delta = s_{(\lfloor \rho|\mathbf{y}|\rfloor)} - s_{(\lfloor \rho|\mathbf{y}|\rfloor+1)}$. If $\varepsilon < \Delta$, the bottom-$\lfloor \rho|\mathbf{y}|\rfloor$ membership is preserved from layer $l$ to $l+1$. Hence, unless many tokens are concentrated near the cutoff, the selected set should remain largely stable across depth.

To investigate whether different layers benefit from different update ratios, we conducted an experiment on GSM8K using LLaDA Instruct with $K_p = 50$, $K_r = 7$, comparing three settings with the same average update budget of 0.25. The results are shown in Table 11. A fixed global $\rho = 0.25$ achieves the highest accuracy, outperforming both a front-heavy allocation (more updates in lower layers) and a back-heavy allocation (more updates in higher layers). While lower-layer updates can propagate to higher-layer features through subsequent Transformer computations, this effect is not strong enough to require a depth-dependent update ratio under the same overall update budget.

*Table 11.* Cross-layer update ratio allocation on GSM8K with LLaDA Instruct. All strategies use the same average budget of 0.25.

| Strategy (Avg Budget = 0.25) | GSM8K Acc. (%) |
|---|---|
| More updates in lower layers ($0.375 \rightarrow 0.125$) | 78.70 |
| More updates in higher layers ($0.125 \rightarrow 0.375$) | 79.23 |
| Fixed Global ($\rho = 0.25$) | **79.83** |

### A.5. Design Choice Between Full V-Cache Overwrite and Selective Update

As discussed in Section 3.2.2, during adaptive partial updates, V-verify computes current Value vectors for all response tokens to score adjacent-step changes. Since this full response-side $\mathbf{V}_r^{\text{new}}$ is already available after the lightweight projection,

we overwrite the entire cached $\mathbf{V}_r$ with $\mathbf{V}_r^{\text{new}}$ rather than only updating the Value vectors of the selected low-similarity tokens. This is an intentional design choice specific to the Value vectors in our V-verify mechanism.

To validate this design, we compared our current approach against a variant that only updates the Value vectors of the selected tokens on GSM8K with LLaDA 8B Instruct. The full overwrite achieves 79.83% accuracy, outperforming the selected-token-only variant at 78.85%. This result favors our current design and confirms that overwriting the full V-cache with the most recent Value vectors provides a better basis for the similarity check in the next step.

### A.6. Impact of Similarity Metric.

We compared cosine similarity and L2 distance as similarity metrics for **V-verify**. On GSM8K with LLaDA 8B Instruct, cosine similarity achieved 78.54% accuracy, significantly outperforming L2 distance at 55.95%. This shows that cosine similarity better captures semantic change, and we adopt it as the default throughout our method.

### A.7. Complexity and Latency Analysis

In this section, we provide a detailed computational complexity analysis for the original dLLM inference process and our proposed dLLM-Cache framework.

**Complexity of the Original dLLM Model.** Standard dLLMs, such as LLaDA and Dream, utilize a multi-layer Transformer architecture with **bidirectional attention**. Text generation is performed over **K** iterative denoising steps, starting from a fully masked sequence. At each step, the model executes a full forward pass over the entire input sequence of length $n$. The per-step computational cost, measured in FLOPs, is dominated by the attention and feed-forward network (FFN) layers:

$$\text{FLOPs}_{\text{step}} = T \cdot (8nd^2 + 4n^2d + 6ndm) \tag{8}$$

where $T$ is the number of Transformer layers, $n$ is the sequence length, $d$ is the hidden dimension size, and $m$ is the intermediate size of the FFN. Note that for SwiGLU-based architectures, *e.g.*, LLaDA, the FFN consists of three matrix multiplications (gate, up, and down projections), contributing $6ndm$ FLOPs.

Consequently, the total inference complexity for a standard dLLM is the per-step cost multiplied by the number of steps $K$:

$$\text{FLOPs}_{\text{dLLM}} = K \cdot T \cdot (8nd^2 + 4n^2d + 6ndm) \tag{9}$$

**Complexity with dLLM-Cache.** dLLM-Cache optimizes this process by caching intermediate states and selectively updating only a fraction of tokens. This partitions the computation into three main types: full refreshes, response-only refreshes, and adaptive partial updates. The total complexity can be approximated as:

$$\begin{aligned}
\text{FLOPs}_{\text{dLLM-Cache}} \approx{}& \frac{K}{K_p} \cdot T \cdot (8nd^2 + 4n^2d + 6ndm) \\
&+ \left(\frac{K}{K_r} - \frac{K}{K_p}\right) \cdot T \cdot (8rd^2 + 4rnd + 6rdm) \\
&+ K \cdot \left(1 - \frac{1}{K_r}\right) \cdot T \cdot (2rd^2 + 8\hat{r}d^2 + 4\hat{r}nd + 6\hat{r}dm)
\end{aligned} \tag{10}$$

where $K_p$ and $K_r$ are the refresh intervals for the prompt and response, respectively; $p$ and $r$ are the prompt and response lengths ($n = p + r$); and $\hat{r} = \rho \cdot r$ is the number of updated response tokens during adaptive steps, with $\rho$ being the adaptive update ratio.

**Computation Savings.** It is a weighted sum of three distinct operational modes. The **first term**, $\frac{K}{K_p} \cdot T \cdot (\dots)$, represents the cost of periodic full refreshes. While its per-instance cost is identical to the original dLLM, its frequency is low, controlled by a large $K_p$(often $\geq 100$).

The **second term**, $(\frac{K}{K_r} - \frac{K}{K_p}) \cdot T \cdot (8rd^2 + 4rnd + 6rdm)$, quantifies the cost of the more frequent response-only refreshes. Here, the significant gain becomes apparent: the quadratic attention term is reduced from $O(n^2)$ to $O(rn)$, as we only compute new query vectors for the response tokens (length $r$) to attend to the full sequence (length $n$). This term represents a middle ground, ensuring the evolving response is updated regularly while leveraging a static cache for the prompt.

The **third and final term**, $K \cdot (1 - \frac{1}{K_r}) \cdot T \cdot (2rd^2 + 8\hat{r}d^2 + 4\hat{r}nd + 6\hat{r}dm)$, acts as the cornerstone of our efficiency. This term captures the adaptive partial updates which constitute the vast majority of the denoising steps. It includes a lightweight overhead of $2rd^2$ for the V-verify mechanism that requires computing Value vectors for all response tokens. Crucially, this small fixed cost is negligible compared to the substantial savings achieved by replacing the full computation over $r$ tokens with a sparse subset defined as $\hat{r} = \rho \cdot r$. With $\rho$ typically set to 25%, this approach significantly reduces the burden on the quadratic attention and heavy FFN layers. Thus, our method trades a minimal verification cost for a massive reduction in the dominant computational loads.

The primary source of acceleration in dLLM-Cache stems from applying the sparse update strategy to both the attention and FFN layers. Specifically, we replace the expensive full-sequence operations with lightweight partial updates: the quadratic attention term is reduced from $4n^2d$ to $4\hat{r}nd$, and the heavy FFN computation is cut from $6ndm$ to $6\hat{r}dm$. The relative computational savings can be expressed as:

$$\text{Savings} = 1 - \frac{\text{FLOPs}_{\text{dLLM-Cache}}}{\text{FLOPs}_{\text{dLLM}}} \tag{11}$$

As demonstrated in our experiments, this significant reduction in computational demand leads to substantial improvements in inference speed, achieving up to a $9.1\times$ FLOPs reduction in practical scenarios.

### A.8. Theoretical Error Bound Analysis

Deriving strict, closed-form error bounds for complex iterative systems such as Transformer-based models remains an open and challenging problem. In this section, we provide a theoretical analysis of the approximation error introduced by dLLM-Cache, and show that it can be bounded under reasonable assumptions, offering insight into the mechanisms that contribute to its empirical stability.

**Error Propagation Formulation.**    Our analysis begins by formalizing the error source. Let $y_k$ be the true hidden state trajectory and $\tilde{y}_k$ be the approximated trajectory. The error at step $k$ is defined as $\delta_k = y_k - \tilde{y}_k$. This error propagates through the Transformer network, $F$, which we assume to be Lipschitz continuous with a constant $L_F$. The denoising update can be abstracted as:

$$y_{k-1} \approx \alpha_k y_k + (1 - \alpha_k)F(y_k) \tag{12}$$

The error at the next step, $\delta_{k-1}$, can then be bounded as follows:

$$\begin{aligned}
\|\delta_{k-1}\| &= \|(\alpha_k y_k + (1 - \alpha_k)F(y_k)) - (\alpha_k \tilde{y}_k + (1 - \alpha_k)F(\tilde{y}_k))\| \\
&\leq \alpha_k \|\delta_k\| + (1 - \alpha_k)\|F(y_k) - F(\tilde{y}_k)\| \\
&\leq \alpha_k \|\delta_k\| + (1 - \alpha_k)L_F\|\delta_k\| \\
&= (\alpha_k + (1 - \alpha_k)L_F)\|\delta_k\|
\end{aligned}$$

Letting $C_k = \alpha_k + (1 - \alpha_k)L_F$, we have $\|\delta_{k-1}\| \leq C_k\|\delta_k\|$. Since $L_F > 1$ for expressive models, it follows that $C_k > 1$, indicating a natural tendency for the error to amplify exponentially without intervention.

**Dual-Mechanism Error Control.**    This is where the dual-mechanism design of dLLM-Cache becomes critical. The first mechanism is the **periodic error reset** enforced by the response refresh interval, $K_r$. At every refresh step $k_0$ (where $k_0$ (mod $K_r$) = 0), the error is effectively reset, *i.e.*, $\|\delta_{k_0}\| \approx 0$. This guarantees that the error accumulation is confined within finite windows of length $K_r$. If error were to accumulate unchecked for $j$ steps from $k_0$, its magnitude would be roughly:

$$\|\delta_{k_0-j}\| \leq \left(\prod_{i=1}^{j} C_{k_0-i+1}\right)\|\delta_{k_0}\| \tag{13}$$

The reset mechanism prevents $j$ from growing indefinitely, thus ensuring the overall error is bounded.

However, to further control the error within these intervals, dLLM-Cache employs the **V-verify adaptive update**. At each non-refresh step, V-verify updates a fraction $\rho$ of the tokens. This means only the error corresponding to the cached $(1 - \rho)$ fraction of tokens, let's denote its projection by $P_{cache}$, is amplified. The error recursion is more accurately described as:

$$\|\delta_{k-1}\| \leq C_k\|P_{cache}(\delta_k)\| + \epsilon_{\text{step}} \tag{14}$$

where $\epsilon_{\text{step}}$ is the small error from the newly computed tokens. Since $\|P_{cache}(\delta_k)\|$ is strictly smaller than $\|\delta_k\|$, this leads to a smaller effective error amplification factor, $C'_k < C_k$. Consequently, the maximum error accumulated within a refresh window is significantly reduced. Instead of peaking at an order of $O(C^{K_r})$, the error peak is now on the order of $O((C')^{K_r})$, which is a much tighter bound.

**Conclusion and Experimental Alignment.** In conclusion, our mathematical analysis demonstrates that the approximation error in dLLM-Cache is robustly bounded. The upper bound $\sup_k |\delta_k|$ depends on the model's properties through $C_k$, on the refresh interval $K_r$, and on the adaptive update ratio $\rho$, which determines $C'_k$. The periodic reset $K_r$ provides a hard guarantee of boundedness by preventing infinite error accumulation, while the V-verify mechanism significantly tightens this bound by actively suppressing error growth at almost every step.

This theoretical finding aligns perfectly with our experimental observations, as shown in Figure 5(b). The configuration without V-verify ($\rho = 0$) exhibits a significant decline in accuracy as $K_r$ increases. In contrast, the configuration with V-verify enabled (even when updating only 25% of tokens, $\rho = 0.25$) maintains nearly the same accuracy as the baseline, even at larger $K_r$ values that yield substantial computational savings. This demonstrates that the considerable acceleration achieved by our method, without sacrificing high fidelity, is attributable to well-controlled errors.

### A.9. Proof of Storage Overhead of Caching

*Theorem: The storage overhead of caching in our method is $O(T \times d \times 4 \times L)$, where $T$ is the number of tokens, $d$ is the embedding dimension, and $L$ is the number of layers.*

*Proof.* For each Transformer layer, dLLM-Cache caches four intermediate feature tensors: **K**, **V**, **AttnOut**, and **FFNOut**. Each tensor has size $T \times d$, so the cache size per layer is

$$M_{\text{layer}} = 4 \times T \times d$$

Across $L$ layers, the total cache contains

$$M_{\text{total}} = L \times M_{\text{layer}} = L \times 4 \times T \times d$$

elements. With bfloat16 storage, this corresponds to $2 \times 4 \times L \times T \times d$ bytes. Therefore, the storage overhead is:

$$O(T \times d \times 4 \times L)$$

This completes the proof. □

### A.10. Limitations

While dLLM-Cache demonstrates significant acceleration with competitive generation quality, it has several limitations that present opportunities for future work. (1) Memory overhead for long sequences. Without memory management systems such as PagedAttention, caching intermediate features increases memory overhead for long-context scenarios relative to standard dLLM inference. Recent works have begun addressing memory and eviction issues in dLLM long-context settings, and integrating these techniques with dLLM-Cache is a promising direction. (2) Approximate nature. Unlike exact KV caching in autoregressive models, dLLM-Cache is an approximation technique that trades a small amount of accuracy for substantial speed gains. While the quality loss is generally minor, as shown in our experiments, it is not fundamentally lossless. We hope that openly acknowledging these limitations will guide future efforts to build on and improve dLLM-Cache.

### A.11. Experimental Details

This section summarizes the experimental configurations for both Base and Instruct variants of the evaluated dLLMs. For each task, we report the number of denoising steps, block length, generation length, remasking strategy, few-shot setting, prompt refresh interval $K_p$, and response refresh interval $K_r$. Unless otherwise specified, all models use low-confidence remasking. The baseline and dLLM-Cache are evaluated under the same PyTorch and Hugging Face Transformers framework. The batch size is 8 for Tables 1 and 2, and 1 for Table 3.

The values of $K_p$ and $K_r$ can be flexibly adjusted according to task requirements rather than through hyperparameter tuning. Smaller intervals can be used for accuracy-sensitive tasks such as code generation or mathematical reasoning, while larger intervals reduce computation when latency is the primary concern. Thus, these settings control the practical balance between precision and efficiency, instead of serving as the source of the observed speedups.

The magnitude of gains sometimes varies across Base and Instruct models due to benchmark configurations from prior work (Nie et al., 2026). For example, MMLU uses a 256-token generation length and decoding steps for Base but only 3 for Instruct, leading to different speedup ratios since our acceleration scales with the number of tokens and denoising steps, as detailed in Appendix A.7.

*Table 12.* Experimental settings for the Instruct model across selected benchmarks.

| Task | Steps | Block Len | Gen Len | Few-shot |
|------|-------|-----------|---------|----------|
| GSM8K | 256 | 8 | 256 | 4 |
| GPQA | 128 | 64 | 128 | 5 |
| Math | 256 | 256 | 256 | 0 |
| MMLU-pro | 256 | 256 | 256 | 0 |
| MMLU | 3 | 3 | 3 | 5 |
| MBPP | 512 | 32 | 512 | 3 |
| BBH | 256 | 256 | 256 | 3 |
| HumanEval | 512 | 32 | 512 | 0 |

*Table 13.* Interval steps for LLaDA Base across selected benchmarks.

| | GSM8K | GPQA | Math | MMLU-pro | MMLU | BBH | MBPP | HumanEval | Avg. |
|------|-------|------|------|----------|------|-----|------|-----------|------|
| $K_p$ | 25 | 100 | 50 | 100 | 100 | 50 | 25 | 100 | 69 |
| $K_r$ | 5 | 8 | 8 | 6 | 6 | 6 | 4 | 5 | 6 |

*Table 14.* Interval steps for LLaDA Instruct across selected benchmarks.

| | GSM8K | GPQA | Math | MMLU-pro | MMLU | BBH | MBPP | HumanEval | Avg. |
|------|-------|------|------|----------|------|-----|------|-----------|------|
| $K_p$ | 50 | 50 | 50 | 51 | 100 | 100 | 100 | 25 | 66 |
| $K_r$ | 7 | 6 | 1 | 3 | 7 | 5 | 5 | 5 | 5 |

*Table 15.* Interval steps for Dream Base across selected benchmarks.

| | GSM8K | GPQA | Math | MMLU-pro | MMLU | BBH | MBPP | HumanEval | Avg. |
|------|-------|------|------|----------|------|-----|------|-----------|------|
| $K_p$ | 100 | 100 | 100 | 25 | 100 | 25 | 25 | 5 | 60 |
| $K_r$ | 8 | 8 | 4 | 2 | 2 | 4 | 8 | 1 | 5 |

*Table 16.* Interval steps for Dream Instruct across selected benchmarks.

| | GSM8K | GPQA | Math | MMLU-pro | MMLU | BBH | MBPP | HumanEval | Avg. |
|------|-------|------|------|----------|------|-----|------|-----------|------|
| $K_p$ | 25 | 10 | 50 | 5 | 100 | 10 | 10 | 50 | 33 |
| $K_r$ | 2 | 8 | 1 | 1 | 8 | 2 | 8 | 1 | 4 |

