# OpenReview forum: "dLLM-Cache: Accelerating Diffusion Large Language Models with Adaptive Caching"
_ICML.cc/2026/Conference — ICML 2026 regular_

### Official Review · Reviewer_3Sc1 · 2026-03-09

**Soundness:** 3
**Presentation:** 3
**Significance:** 2
**Originality:** 2
**Overall Recommendation:** 4
**Confidence:** 4

**Summary:**

The paper proposes a training-free caching method based on the similarity of K/V features across neighboring diffusion steps. Different from existing cache mechanisms for diffusion language models, the paper caches all K/V features at initialization and then decides whether to refresh cached features based on V feature similarity across neighboring diffusion steps under a fixed refresh budget. This bold cache design pushes the inference speed of diffusion language models to a new level, while still preserving model performance reasonably well.

Overall, although the core novelty is limited, I find the paper practically useful.

**Compliance With Llm Reviewing Policy:**

Affirmed.

**Key Questions For Authors:**

1. Is the selected token set expected to remain stable across layers, or can it change significantly with depth?

2. Have the authors studied whether lower-layer recomputation amplifies higher-layer changes, and whether a fixed global update ratio ρ remains appropriate throughout the network depth?

**Limitations:**

Yes

**Strengths And Weaknesses:**

Strength

1. Clear practical value. The proposed cache mechanism is simple but effective. The speedups are substantial, while model performance is largely preserved.

2. Good supplementary analysis. The paper provides a comparison with other cached diffusion language models in the complexity analysis.

Weakness:
1. Limited methodological novelty. The improvement comes more from system design than from a fundamentally new insight.

2. The cross-layer update behavior is under-specified. The method performs token selection in a layer-wise manner, but the design choice seems somewhat arbitrary. The paper does not clearly explain the reason for this design, and it also lacks an investigation of how lower-layer refreshes affect higher layers, both in terms of refresh budget and token selection.

---

> ### Author Rebuttal · Authors · 2026-03-30
>
> Dear Reviewer 3Sc1,
>
> Many thanks to your valuable comments and questions, which help us a lot to improve our work. We address your questions as follows.
>
> ---
>
> > [W1] The contribution appears closer to system design than to a fundamentally new methodological insight.
>
> [A1] We appreciate the feedback regarding our methodological contribution. While the system-level design is central to our work, these engineering choices stem directly from our quantitative observation of **adjacent step feature similarity within bidirectional attention diffusion models**. Our method relies on **empirical data** rather than heuristics. For instance, Figure 1 demonstrates that prompt features remain highly stable across denoising steps, whereas response features evolve dynamically. This observation justifies our use of a long cache interval for the prompt and a more frequent, adaptive interval for the response. Furthermore, the variation in the Value vector shows a strong positive correlation with the variation in subsequent AttnOut and FFNOut blocks, as shown in Figure 2. This validates V-verify as an effective, low-overhead proxy for identifying dynamic tokens. The selected 25% update ratio $\rho$ is similarly supported by the ablation study in Figure 5, which identifies the optimal trade-off between computational savings and quality degradation. We view this work as a baseline demonstrating that significant redundancy exists in dLLMs. This observation facilitates future research into more advanced algorithms for redundant token identification. We will revise the introduction to better highlight these insights.
>
> ---
>
> > [W2 Q1 Q2] The layer-wise token-selection design is under-specified, and the paper does not explain why this choice is made or how lower-layer refreshes affect higher layers.
>
> [A2] We appreciate the chance to clarify the cross-layer behavior of dLLM-Cache. The layer-wise token selection is not arbitrary. Cached features are stored separately for each Transformer layer $l$. At each layer, we cache $\mathbf{K}^{(l)}$, $\mathbf{V}^{(l)}$, $\mathbf{AttnOut}^{(l)}$, and $\mathbf{FFNOut}^{(l)}$ in the Prompt Cache $\mathcal{C}_p$ and the Response Cache $\mathcal{C}_r$. The V-verify score is therefore defined locally at layer $l$ by comparing the current Value vector with its cached version, which makes per-layer token selection **the most consistent choice for the cache design**.
>
> The selected token set is expected to remain largely **stable** across adjacent layers. Let $s_j^{(l)}$ denote the V-verify score of response token $j$ at layer $l$. Under standard smoothness assumptions, if the layer map and the Value projection are Lipschitz, then $|s_j^{(l+1)}-s_j^{(l)}|\le\varepsilon_l$. Let $\Delta_l=s_{(\rho r+1)}^{(l)}-s_{(\rho r)}^{(l)}$ denote the margin at the selection boundary, where $r$ is the number of response tokens. If $\Delta_l>2\varepsilon_l$, the bottom-$\rho$ membership is preserved from layer $l$ to $l+1$. Hence, unless many tokens are concentrated near the cutoff, the selected set should remain largely stable across depth.
>
> This reasoning is also why we use a fixed global update ratio $\rho$. In the paper, we already studied the effect of the adaptive update ratio $\rho$, and Figure 5 shows that value-based selection achieves the highest accuracy around $\rho\approx$ 0.25. Our theoretical analysis shows that the amplification of approximation error is controlled by the response refresh interval $K_r$ and the adaptive update ratio $\rho$, and that V-verify reduces the effective error amplification factor by updating a fraction $\rho$ of the response tokens. Since lower-layer updates can propagate to higher-layer features through subsequent Transformer computations, we also examined whether different layers should use different $\rho$ values. On GSM8K using LLaDA Instruct with $K_p$ = 50 and $K_r$ = 7, we find that **a fixed global $\rho$ remains the best** configuration among three settings with the **same average update ratio**:
>
> | Strategy (Avg Budget = 0.25)                              | GSM8K Acc. |
> | --------------------------------------------------------- | ---------- |
> | More updates in lower layers (0.375 $\rightarrow$ 0.125)  | 78.70      |
> | More updates in higher layers (0.125 $\rightarrow$ 0.375) | 79.23      |
> | Fixed Global ($\rho$ = 0.25)                              | **79.83**  |
>
> Thus, while lower-layer recomputation can influence higher-layer features, our evidence does not suggest that this effect is strong enough to require a depth-dependent update ratio. Assigning more updates to lower layers does not outperform the fixed global $\rho$ under the same overall update budget. We will include the cross-layer analysis presented above in the revised manuscript, and we are committed to further exploring V-verify in future work.

---

> > ### Author Rebuttal · Reviewer_3Sc1 · 2026-04-03
> >
> > Thank you for your rebuttal. I think it resolved my questions and I will maintain my positive score,

---

> > > ### Author Response · Authors · 2026-04-05
> > >
> > > Thank you for your valuable feedback. We are glad to have addressed your concerns and will incorporate your suggestions. We deeply appreciate your time and contribution to improving our paper.

---

### Official Review · Reviewer_DAEj · 2026-03-10

**Soundness:** 3
**Presentation:** 3
**Significance:** 3
**Originality:** 2
**Overall Recommendation:** 4
**Confidence:** 4

**Summary:**

This work proposes a caching mechanism for diffusion LLMs. The central insight lies in the fact that there are two major redundancies when performing uniform computations on all tokens at each step. dLLM-Cache proposes to only periodically update the prompt cache and selectively update a subset of tokens’ cache for the response tokens. The result is a training-free method that achieves up to 9x speedup on Llada with minimal loss.

**Compliance With Llm Reviewing Policy:**

Affirmed.

**Key Questions For Authors:**

Please see the weakness part.

**Limitations:**

No explicit section on limitation. Please refer to the weakness part.

**Strengths And Weaknesses:**

**Strength**
1. The problem of developing an efficient cache mechanism for diffusion LLMs is an important question. This paper proposes a novel method that builds upon several key observations.
2. Figure 1 is a good example showing across-step similarities for prompt and response tokens.
3. Figure 2 is a good illustration that motivates the V-verify mechanism for dynamically decoding what tokens to cache for the response part.
Two main pure diffusion models are evaluated across many tasks, making the overall evaluation quite comprehensive.

**Weakness**
1. There are already many works that try to solve this problem and share some of the same motivations (this should be put in the main related work instead of having a discussion in the appendix):
* https://arxiv.org/abs/2505.15781
* https://arxiv.org/abs/2505.22618
* https://arxiv.org/abs/2509.26328
* https://arxiv.org/abs/2503.09573 (block diffusion variants). BD is exact in terms of quality.
2. No batched behavior is mentioned in the experiments. For the response tokens, when the number of active tokens differs across samples, can dLLM-Cache efficiently handle it?
3. The implementation framework is not mentioned. Efficiency claims need to be made in the context of the framework (is it huggingface, pure pytorch, SGLang, and do we use cuda-graph or torch compile for the baseline)?

---

> ### Author Rebuttal · Authors · 2026-03-30
>
> Dear Reviewer DAEj,
>
> Many thanks to your valuable comments and questions, which help us a lot to improve our work. We address your questions as follows.
>
> ---
>
> > [W1] Related work on similar acceleration ideas should be discussed in the main paper rather than mostly in the appendix.
>
> [A1] We fully agree this. In the revision, we will move the discussion of closely related work such as dKV-Cache, Fast-dLLM, and other dLLM acceleration approaches into the main related work section.
>
> ---
>
> > [W2] The experiments do not discuss batched behavior. If different samples have different numbers of active tokens, can dLLM-Cache still run efficiently?
>
> [A2] Thank you for pointing this out. As noted in Appendix A.9, the experiments reported in Tables 1 and 2 were evaluated in batches with batch size 8. dLLM-Cache **remains efficient** even when samples in the same batch have different numbers of active tokens. This is a common issue in batched dLLM inference and does not affect the cache reuse mechanism. In practice, it can be handled with padding or with variable-length operators such as *flashattention_varlength*.
>
> To further examine this case, we ran additional experiments with LLaDA Instruct on GSM8K using a single A100 80G SMX GPU. When *flashattention_varlength* is used to handle unequal numbers of active tokens, batch decoding with batch size 4 reaches **61.74** TPS, compared with 19.36 TPS for batch size 1. These results show that our method **remains efficient in batch settings with dynamic active tokens**.
>
> ---
>
> > [W3] The implementation framework is not specified, so the efficiency claim lacks context.
>
> [A3] We thank the reviewer for focusing on the system implementation. To isolate the contribution of the algorithm, we implement the baseline in standard PyTorch with Hugging Face Transformers, without specialized inference engines or compiler-level optimizations. Both the baseline and dLLM-Cache are evaluated under the same framework to ensure a fair comparison.
>
> To support the dynamic refresh strategy in dLLM-Cache, we add a multi-branch **CUDA Graph** mechanism at the Transformer-layer level. The system maintains four pre-recorded graphs corresponding to the four execution paths defined in our algorithm: Full Refresh, Refresh Prompt Only, Refresh Response Only, and Adaptive Update, where the last is invoked most frequently. At runtime, each denoising step is dispatched to the appropriate graph with negligible overhead. After excluding the one-time graph recording cost, this multi-branch CUDA Graph mechanism provides an additional **1.51×** speedup over the baseline implementation. These results support the effectiveness of our design.

---

> > ### Author Rebuttal · Reviewer_DAEj · 2026-04-02
> >
> > Thank you for the thorough rebuttal. My main concerns — particularly regarding W2 and W3 — have been adequately addressed. Would love to see it implemented in SGLang or vLLM as a future work, which would be very useful for the community. I will maintain my positive score.

---

> > > ### Author Response · Authors · 2026-04-05
> > >
> > > Thank you for the positive evaluation. We are glad our rebuttal addressed your concerns. We fully embrace your suggestions and greatly appreciate your time and effort in improving this paper.

---

### Official Review · Reviewer_sV7Y · 2026-03-12

**Soundness:** 3
**Presentation:** 3
**Significance:** 2
**Originality:** 3
**Overall Recommendation:** 4
**Confidence:** 4

**Summary:**

The authors observe that dLLM inference involves a static prompt and a partially dynamic response, where most tokens remain stable across adjacent denoising steps. Based on this observation, they propose a heuristic method called dLLM-Cache. The method combines long-interval prompt caching with partial response updates guided by feature similarity. This allows the model to reuse cached computations and update only a subset of tokens each step. As a result, it achieves significant speedup over standard inference without compromising output quality.

**Compliance With Llm Reviewing Policy:**

Affirmed.

**Final Justification:**

The authors’ response partially addresses my concerns, but does not fully resolve the main issues. Therefore, I maintain my original evaluation.

**Key Questions For Authors:**

1.The paper repeatedly states that the maximum speedup reaches 9.1×, but I could not find any speedup ≥9× in the tables or figures of the paper. I would like to know how this 9.1× number is obtained.

2.The paper should compare with other related inference acceleration methods, or provide evidence showing whether the proposed method is compatible with them.

**Limitations:**

The practical limitations mainly lie in how the many hyperparameter settings affect the actual speedup in practice, as well as the compatibility with other acceleration methods. The latter is not discussed in the paper.

**Strengths And Weaknesses:**

Strengths:

1.The idea is reasonable, and the heuristic design of the method appears to be carefully constructed.

2.Although there have been several related works, this paper is clearly distinguishable from other dLLM inference acceleration methods, such as [1,2].

[1] dKV-Cache: The Cache for Diffusion Language Models. NeurIPS 2025.

[2] FlashDLM: Accelerating Diffusion Language Model Inference via Efficient KV Caching and Guided Diffusion. ICLR 2026.

3.The experimental results show a substantial improvement in inference speed, and the acceleration effect is observed across different models and datasets.

Weaknesses:

1.Stability of the method. In the experiments reported in Table 2 and Table 3, the speedup on the TPS metric ranges from 1.28× to 5.26×, which shows quite large variation. Since dLLM-Cache is a heuristic approach and involves several hyperparameters, the acceleration effect can fluctuate noticeably.

2.Lack of comparison with related methods. There are already many works on accelerating dLLM inference. However, the paper does not compare its method with these existing approaches, nor does it show whether they can be combined with dLLM-Cache.

3.The writing can still be improved. In particular, Figure 3 is quite large, and the text inside the figure has inconsistent font sizes. Some of the text is even much larger than the main body text. It is recommended to adjust and revise the figure.

---

> ### Author Rebuttal · Authors · 2026-03-30
>
> Dear Reviewer sV7Y,
>
> Many thanks to your valuable comments and questions, which help us a lot to improve our work. We address your questions as follows.
>
> ---
> > [W1] The speedup varies substantially across tasks, so the stability of the heuristic needs better explanation.
>
> [A1] We appreciate the suggestion to clarify the TPS variation. This variance **does not result from unstable hyperparameter tuning across tasks**. Our main experiments consistently use a fixed adaptive update ratio $\rho$ = 0.25, and Figures 4-6 demonstrate that our method is reasonably robust. Instead, the varying speedup ratios stem from differences in the **total number of tokens and decoding steps across benchmarks** (see Appendix A.6). Because we follow the original LLaDA and Dream evaluation protocols, task lengths differ considerably. For instance, evaluating MMLU with LLaDA Instruct requires only 3 generation tokens or denoising steps, leaving less redundant computation to amortize than longer tasks like GSM8K. **Under fixed configurations, the speedup remains stable**; the observed variation reflects the amount of reusable computation inherent to each benchmark. We will detail this mechanism in the revision.
>
> ---
> > [W2 Q2] The paper lacks comparison with related acceleration methods and does not clearly show compatibility with them.
>
> [A2] We agree that these discussions belong in the main text. Although the current submission includes this evidence, it is located in the appendices. Appendix A.1 (Table 4) shows that dLLM-Cache can be combined with SlowFast Sampling for further throughput gains. Additionally, Appendix A.4 (Table 9) evaluates dLLM-Cache against concurrent acceleration methods, such as dKV-Cache and Fast-dLLM, on both LLaDA and Dream. For your convenience, we summarize the representative results below.
>
> Compatibility with SlowFast Sampling (Appendix A.1 / Table 4):
>
> | **Task** | **Method**           | **TPS↑**  | **Speed(TPS)↑** | **Score↑** |
> | ---- | -- | -- | ----- | --- |
> | GSM8K    | LLaDA                | 4.55      | 1.00×           | 69.83      |
> |          | **+Cache +Sampling** | **26.99** | **5.93×**       | **69.60**  |
> | GPQA     | LLaDA                | 3.31      | 1.00×           | 31.47      |
> |          | **+Cache +Sampling** | **29.06** | **8.78×**       | **33.48**  |
> | BBH      | LLaDA                | 4.04      | 1.00×           | 44.97      |
> |          | **+Cache +Sampling** | **36.04** | **8.92×**       | **44.81**  |
>
> Comparison with concurrent methods (Appendix A.4 / Table 9):
>
> | **Task**  | **Method**              | **TPS↑**  | **Speed↑** | **Memory↓** | **Score↑** |
> | :-------- | :---------------------- | :-------- | :--------- | :---------- | :--------- |
> | MMLU      | LLaDA Instruct          | 10.12     | 1.00×      | 15.54       | 61.24      |
> |           | + dKV-Cache             | 14.34     | 1.42×      | 17.88       | 60.87      |
> |           | + Fast-dLLM             | 20.51     | 2.03×      | 17.13       | 61.43      |
> |           | **+ dLLM-Cache (Ours)** | **21.23** | **2.10×**  | **16.61**   | **62.82**  |
> | HumanEval | LLaDA Instruct          | 10.57     | 1.00×      | 15.39       | 38.71      |
> |           | + dKV-Cache             | 14.40     | 1.36×      | 17.17       | 37.20      |
> |           | + Fast-dLLM             | 21.50     | 2.03×      | **16.60**   | 36.59      |
> |           | **+ dLLM-Cache (Ours)** | **44.77** | **4.24×**  | 16.65       | **39.02**  |
>
> In the revision, we will move these results into the main related work and experiment sections so the compatibility and comparative advantages are immediately visible.
>
> ---
>
> > [W3] Figure 3 should be improved visually.
>
> [A3] Thank you for your suggestion. We will revise Figure 3 accordingly. In particular, we will reduce its size, make the internal font sizes consistent, and align its typography more closely with the main body text.
>
> ---
>
> > [Q1] The paper repeatedly mentions a maximum 9.1x speedup, but this number is not easy to locate in the main paper.
>
> [A4] Thank you for catching this. The reported 9.1× speedup refers to the FLOPs reduction achieved on the long-context benchmark LongBench-HotpotQA, as detailed in Appendix A.2. To avoid confusion, we will revise the Abstract and Introduction to clarify this as: _"...achieves up to 9.1× FLOPs speedup on LongBench..."_

---

> > ### Author Rebuttal · Reviewer_sV7Y · 2026-04-04
> >
> > Thank you for your thoughtful response. I have carefully considered your rebuttal. However, I will maintain my original score.

---

> > > ### Author Response · Authors · 2026-04-05
> > >
> > > We sincerely appreciate the time you took to read our response. Thank you again for your valuable suggestions and for helping us enhance the quality of our paper.

---

### Official Review · Reviewer_nErT · 2026-03-15

**Soundness:** 3
**Presentation:** 3
**Significance:** 4
**Originality:** 4
**Overall Recommendation:** 5
**Confidence:** 4

**Summary:**

Diffusion LLMs are a promising alternative to autoregressive LLMs. However, a major limitation has been the inefficiency of inference due to the inability to have a fixed KV cache. In this work, the authors propose a methodology, dLLM-Cache, to still build a cache by only recomputing the entries for the tokens for which the value matrix suggests the most variation across timesteps, and only perform full cache recomputations sparsely. This is motivated by the fact that the latent representations of a large amount of tokens do not change significantly across timesteps. The authors proceed to show that their method leads to significant speedups while remaining competitive in accuracy on the evaluations they considered, compared to the model not using dLLM-Cache.

**Compliance With Llm Reviewing Policy:**

Affirmed.

**Final Justification:**

The authors promised to make all the changes we discussed to resolve the weaknesses and typos mentioned in my review, so I raise my score to a 5 as promised. I personally think this paper does a good job of proposing and evaluating a promising method that, even if not perfect, will be of interest to the community and possibly lead to more research directions and refinements. I honestly hesitated to put a 6 based on the potential impact this could have for serving dLLM, but it says "flawless" and there are rather minor flaws (as there always will be with very empirical work). But to the AC, if the question is "would I spend money to try this in prod", I would answer yes for anyone working on dLLM, so I believe this should clearly be accepted based on that, if only to make the dLLM community think on this.


Now, dear AC, forgive the rant, but someone had the great idea to change openreview AGAIN and prevent me from properly engaging in a discussion with the authors by not allowing replies past the rebuttal acknowledgement that can be read by paper authors... So I'll engage with the reply to the rebuttal acknowledgement here, hoping next conferences will resolve this new nonsense. I really marvel at the thought processes involved when denying the ability to properly engage in more than 1 message of discussion, when our goal as program committee is not just to be binary classifiers, but to ensure resolving all issues so that papers of interest to the community get published with the highest possible quality. Authors luckily were diligent and followed up with edits of their own.

**Key Questions For Authors:**

**Questions**
1. So essentially, you still have to recompute the full value matrix of the response tokens every step for the adaptive partial update, correct? Do you keep it? If not, why not?
2. Did you experiment with forcing the recomputation of (or alternatively, keeping) the K and V of tokens that got unmasked or changed in the previous timestep? Or perhaps it would be too expensive since there would be way too much change every timestep.
3. More of a remark than a question: be careful about the Llama 3 comparisons: the datasets used for LLaDA and Llama 3 are not the same, so I would be careful about any comparison to Llama 3. It’s ok in tech reports, but a scientific paper needs to at least clearly indicate this caveat if it’s going to make a comparison on the basis of a fixed size of model, varying architecture. In my experience working on the LLM stack, data tends to make the most difference when it comes to LM evals. However, readers less experienced with training LLMs might misunderstand that architecture is the major factor of that gap. I’m not saying you should remove Llama 3 though, because it is of interest to have the comparison e.g. of speed or memory utilisation.


**Suggestions**
1. Eq 5: the notation $\hat{y}^{(0)}$ probably is missing a $k$ sub/superscript, no? It is the denoised estimation, but at a specific step, so if I’m correct, I recommend indicating it more explicitly.
2. p3: “The specific strategy for S may involve confidence-based remasking or semi-autoregressive block updates.” please at least cite references so readers unfamiliar with this literature have pointers, since I expect your readership to include some people unfamiliar with dLLMs.


**Typos**
1. p4: “only a small fraction of response tokens change” should be “changes” since “a small fraction” is singular.
2. Figure 3: “V Verify” should be “V-verify” to remain consistent with the name you have been using in the rest of the paper. “$V_{K-1}$” in the cosine similarity part should have its $K$ italicised. Also, is there any reason why the box for Q on the middle part of the figure is smaller than on the left part, or than K and V?
3. (throughout the paper) “LLaMa 3” should be “Llama 3”. After the first Llama version, Meta decided to uppercase the first letter, lowercase all others.
4. Figure 4: “gary” instead of “gray” in the caption.

**Limitations:**

No, see weaknesses.

**Strengths And Weaknesses:**

**Strengths**
1. I believe diffusion language models are a topic of significant interest to the community. It is true that the inability to build efficient strategies for memory and latency such as KV caches has limited their use. A paper proposing a cache for dLLMs is certainly going to be relevant to many researchers.
2. The paper is overall generally well-written and easy to follow.
3. Their method is intuitive and is shown to work well on their experiments. I appreciate the FLOPs estimation and memory comparison. It’s also particularly nice that it is shown to work even better in conjunction with other existing methods.


**Weaknesses**

Overall, I believe the paper is interesting and I’d be happy to increase my score if the following issues were addressed:
1. There are a few presentation issues to solve. I expand on them in the rest of my review in more details about weaknesses, typos and suggestions (and in particular, the easily-fixed “lossless” overclaim, see more details #5).
2. Your method has limitations. It’s ok! It already opens a door for the research community to iterate on solving them, so I’m not penalising you in my review for not addressing all of them, but please at the very least discuss them and be explicit about them. See more details.


**More details about weaknesses**
1. Figure 1: please add the relevant x-axis legend below the subfigures. Above is typically for figure title if there is no clear axis above. For example, you can leave prompt and response above, but below indicate token index / position.
2. Figure 2: it’s guessable (and confirmed reading the text of course) that you’re evaluating correlations of the cosine similarities through time of (K, V) and the cosine similarities through time of (AttnOut, FFNOut). For legibility’s sake though, I think you probably want to improve the caption so there is no room for doubt from the first read. I suspect many people (and in particular for LLM papers, because there are so many) first tend to skim through a paper before deciding to give it a more careful read, so I’m sure you understand why it’s best to be explicit here. I’m thinking specifically of “compute the correlation between their similarity with (a) and (c) AttnOut” which I guessed correctly I believe, in spite of the inexact phrasing, which currently says “correlation between the similarity of <K or V through steps> , and the output of AttnOut”.
3. Figure 4b: the caption is clear. However, the indication of the hyperparameters for 3 out of the 4 curves is slightly confusing. Because there are 3 colours, the first reflex is to associate one of the 3 written combo of parameters with a given colour. Again, not a big deal since the caption is clear, but I suggest making a minor modification to avoid that initial confusion.
4. Figure 4c: this figure is too confusing. You do not have a lot of real estate to have that big red text and logos. I like your paper! But it’s also odd to see that in a scientific paper. You also have been varying $K_p$ and $\rho$ in the other subfigures so please indicate which value you chose there to ensure clarity. Finally, I suspect many readers might not immediately get that this graph suddenly decided to indicate the curves of varying $K$ WITHOUT dLLM-cache, indicating in a single point in grey the performance WITH dLLM-cache. I recommend clarifying that in the caption! You can also add your x2 acc and x5 faster improvements in the caption instead.
5. “lossless impact on response quality” is a big claim that is not substantiated by the more detailed evals in the tables, which suggest instead that you win some, you lose some. Of course, in my experience, any variation within up to a percent of evals tends to be due to stochasticity, but there are a few evals where your deviation is larger than that for wins or losses. There likely are other points in the paper where this is made too generally. My advice would be to use the word “competitive”.
6. There are no discussions of limitations, yet your method has several limitations. For example, it will exacerbate dLLM’s out-of-memory issues if we increase the sequence length since there is no paged attention. The end of the discussion gets close to the memory boundedness of having to always recheck the whole V matrix (see my question 1, I might be getting that wrong). The method is still an approximation but is not fundamentally lossless, unlike KV caches. It introduced new hyperparameters that will have to be optimised by users. Etc.

---

> ### Author Rebuttal · Authors · 2026-03-30
>
> Dear Reviewer nErT,
>
> Many thanks to your valuable comments and questions, which help us a lot to improve our work. We address your questions as follows.
>
> ---
> > [W1] The presentation still has several fixable issues, especially in the figures and captions.
>
> [A1] Thank you very much. We have improved [Figure 1](https://anonymous.4open.science/r/rf4m9x2c-FB81/Figure1.png) by adding clearer x-axis labels, rewritten the caption of [Figure 2](https://anonymous.4open.science/r/rf4m9x2c-FB81/Figure2.png), and revised [Figure 4(b,c)](https://anonymous.4open.science/r/rf4m9x2c-FB81/Figure4.png), as shown in the updated figures linked above. We have also fixed the notation issue in Eq 5 by revising $\hat{\mathbf{y}}^{(0)}$ to $\hat{\mathbf{y}}^{(0\mid k)}$, added the missing citation on p3, and corrected the listed typos. We appreciate this detailed feedback, which has helped us improve the readability of the paper.
>
> ---
> > [W2] "Lossless impact on response quality" is too strong and should be stated more cautiously.
>
> [A2] We deeply appreciate it. To avoid overclaiming, we will replace "lossless" with more cautious wording such as "competitive" in the revised manuscript.
>
> ---
> > [W3] The method has real limitations, but the paper does not discuss them sufficiently.
>
> [A3] We agree and will add a dedicated limitations section. Without memory management systems such as PagedAttention, caching these additional features increases memory overhead for long sequences relative to standard inference. Recent works such as Sparse-dLLM [1], MaskKV [2], and d$^2$Cache [3] have begun addressing memory and eviction issues in long-context settings. In addition, dLLM-Cache is approximate rather than strictly lossless and introduces extra hyperparameters. To reduce this burden, we will **provide default configurations**, supported by Figures 4–6, together with practical guidelines in the future code release. Regarding the concern raised in Q1, the method compares the current V of response tokens only with the cached V from the immediately preceding denoising step, and this cache is **overwritten at each step**, so **memory does not grow** with the number of denoising steps.
>
> [1] Sparse-dLLM: Accelerating Diffusion LLMs with Dynamic Cache Eviction. arXiv 2025.
>
> [2] Mask Tokens as Prophet: Fine-Grained Cache Eviction for Efficient dLLM Inference. arXiv 2025.
>
> [3] d$^2$Cache: Accelerating Diffusion-Based LLMs via Dual Adaptive Caching. ICLR 2026.
>
> ---
> > [Q1] In adaptive partial update, do you still recompute the full Value matrix of response tokens at every step? Do you keep it?
>
> [A4] Yes. In adaptive partial update, we compute the Value vectors of all response tokens at each denoising step. V-verify compares the current V with the cached V from the previous denoising step, uses the similarity scores to select tokens for update, and overwrites the cache with the current V for the next step. The comparison is thus **only between adjacent denoising steps**. We do not keep a longer history of full V matrices, so memory does not accumulate over time. We will clarify this workflow in the main text and point readers more clearly to Algorithm 6.
>
> ---
> > [Q2] Did you test forcing recomputation, or alternatively keeping, the K/V of tokens that were changed or unmasked in the previous step?
>
> [A5] We thank the reviewer for this suggestion. We ran an ablation to test forced recomputation around the early unmasking stage. Specifically, we force a full refresh of response features at a chosen early step and then apply a long cache interval. Results on GSM8K with LLaDA Base are:
>
> |Strategy|step|Accuracy|
> |-|-|-|
> |Forced Recomputation|1|64.75|
> ||2|65.13|
> ||3|65.66|
> ||4|67.55|
> ||5|66.79|
> |**Standard dLLM-Cache**|-| **67.39** |
>
> Since forced recomputation shows no consistent gain, we keep our standard adaptive strategy.
>
> ---
> > [Q3] Comparisons with Llama 3 should be presented more carefully because of training-data differences.
>
> [A6] We agree. In the revision, we will clarify that our primary evaluation focuses on comparing the original dLLM with dLLM-Cache, while the Llama 3 comparison is included only as a broad efficiency reference rather than a direct baseline.
>
> ---
> > [T2] Is there any reason why the box for Q on the middle part of the figure is smaller than on the left part, or than K and V?
>
> [A7] The smaller Q box in the middle panel is intentional and reflects the **adaptive partial update** design. The left panel shows full computation, where Q, K, and V are produced for the entire sequence. In the middle panel, we compute the current V for all response tokens during V-verify to identify low-similarity tokens, and then **recompute Q and K only for the selected response tokens**. Attention is then computed with these selected queries together with the full reused prompt cache and updated response cache. The Q box is therefore smaller because **only a subset of response tokens issues new queries**, while K and V still represent the full context.

---

> > ### Author Rebuttal · Reviewer_nErT · 2026-04-02
> >
> > Thank you for your thorough rebuttal! I really appreciate the anonymised link to the new figures.
> >
> >
> > > W1
> >
> > Fig 4c: Please excuse me if I missed something here. But I don't see a $K$ for which dLLM-Cache is *simultaneously* 5x faster and 2x more accurate as the caption suggests. The 2x more accurate happens at a $K$ where dLLM-Cache is just as fast. The 5x faster happens at a $K$ where it is just about as accurate. So the language should say exactly that, better pareto optimum so "5x faster at similar accuracy".
> >
> > > Q1
> >
> > Sorry I should have clarified my question. Suppose there are two tokens in the sequence, so $V$ can be written as the concat of the value vectors of these two tokens. So you have 4 value vectors, corresponding to the new step's values and the previous step's values. Suppose the second token is flagged for the update. Do you still update $V_{old}$ with BOTH of $V_{current}$'s value vectors (so effectively, assigning the current $V$ to the old one) or do you only update the value vector of the token selected in $V_{old}$? If the logic for $V$ mirrors that of the other quantities, you'd only update the value vectors of the tokens that got flagged but not the others. Hence my questions.
> >
> > > T2
> >
> > Understood. But then why is the blue (recomputation) part of $Q, K, V$ different from one another (when looking at the blue area on the figure)? Also, please still address the other issues mentioned (no need to send the anonymised updated figure).
> >
> > ___
> >
> > **Edit post "reply to rebuttal acknowledgement":** someone had the great idea to change openreview AGAIN and prevent me from properly engaging in a discussion with the authors by not allowing replies past the rebuttal acknowledgement that can be read by paper authors... So I'll engage with the reply to the rebuttal acknowledgement here, hoping next conferences will resolve this new nonsense.
> >
> > Authors, please forgive the ramble.
> >
> > Thank you for the ongoing discussion!
> >
> > > Response to Q1
> >
> > I'm curious, because technically this means information from the updated tokens will force changes into the V even of unupdated tokens. Perhaps it's better this way than freezing the V of unupdated tokens in the cache. Was wondering about that in my original review's question. I don't expect you to run an experiment on this, but please indicate that clearly in the paper (could be in the appendix, as you wish) as there's a subtle design decision here.
> >
> > In the meantime, I raise my score to a 5 in anticipation of you implementing all the changes we discussed, as promised.
> >
> > ___
> >
> > **Edit 2: Response to Follow-up on Q1:** Thank you for running this! I believe adding this somewhere in the paper appendix will be a great addition for any reader who finds themselves wondering about that too.
> >
> > Thank you for engaging in this discussion. Like I said, I believe the paper deserves acceptance. My final piece of advice, independently of this review process, is that the most impactful LLM papers often aren't published; their success comes from being easily usable. Please consider if it makes sense to e.g. make a PR to vLLM and such commonly used libraries for serving.

---

> > > ### Author Response · Authors · 2026-04-03
> > >
> > > We sincerely thank the reviewer for taking the time to carefully read our rebuttal and for providing further insightful questions. We are glad to have the opportunity to clarify these points.
> > >
> > > ---
> > >
> > > **Response to W1**: We fully agree with the reviewer. As stated in Section 5 (Line 408), the 5$\times$ speedup refers to dLLM-Cache at $K$=256 steps compared to the baseline at the same number of steps, while maintaining competitive performance. The 2$\times$ improvement in accuracy is derived from comparing dLLM-Cache at 256 steps against a baseline under an equivalent FLOP budget. We will revise the caption of Figure 4(c) in our manuscript.
> > >
> > > ---
> > >
> > > **Response to Q1**: Thank you for the opportunity to clarify this algorithmic detail. We update the entire $V_{old}$ cache with the newly computed $V_{current}$ for **all response tokens**, rather than only the flagged ones. This is an intentional design specific to the Value vectors in our V-verify mechanism. Because V-verify uses the current Value vectors of all response tokens to compute cosine similarity with those from the previous step, we need to compute $V_{current}$ for the full response sequence. This step is lightweight because it only requires a single linear projection. Since the full $V_{current}$ is already available, we overwrite the entire $V_{old}$ cache with it (Algorithm 6, Line 23). This keeps the similarity check in the next step based on the most recent features **without introducing additional overhead**. We will add a clarifying sentence in Section 3.2.2 in our manuscript to avoid future confusion.
> > >
> > > ---
> > >
> > > **Response to T2:** Thank you for pointing out this issue in Figure 3. The difference in the computed regions arises from our V-verify mechanism. To determine which tokens require updates, **Value vectors are computed for the entire response sequence**, making V's computed (blue) region the **largest**. In contrast, Q and K are recomputed only for the selected tokens. In the algorithm, Q and K are recomputed for the exact same tokens. The slight mismatch in the original figure was simply a drawing artifact to ensure the "Q" label was clearly visible. We apologize for the confusion. We will correct Figure 3 to show identical regions for Q and K.
> > >
> > > We will carefully address all the other issues you mentioned in the revised manuscript.
> > >
> > > ---
> > >
> > > **Response to Follow-up on Q1**
> > >
> > > Thank you very much for this helpful follow-up and for the positive reassessment. We ran an additional experiment on GSM8K with LLaDA 8B Instruct, comparing our current design with a variant that only updates the Value vectors of the selected tokens. The results are **79.83** for our current design and 78.85 for the selected-token-only variant. This favors our current design, and we will clarify this subtle design choice explicitly in Section 3.2.2 and Algorithm 6.
> > >
> > > Once again, we are truly grateful for the time and effort you dedicated to reviewing our paper. We will carefully revise the manuscript to fully incorporate your valuable suggestions.

---

### Decision · Program_Chairs · 2026-04-30

**Decision:**

Accept (regular)

**Comment:**

This paper introduces dLLM-Cache, a training-free adaptive caching framework designed to reduce the high inference latency of diffusion-based Large Language Models (dLLMs). The reviewers appreciated the method's practicality and intuitive design choices, which leverages the stability of prompt and response tokens across denoising steps. The authors provided a detailed rebuttal, clarifying the "lossless" vs. "competitive" performance claims and adding comparisons with concurrent acceleration methods like Fast-dLLM and dKV-Cache. Concerns regarding layer-wise updates and batched efficiency were effectively addressed through additional experiments and technical justifications, including the implementation of CUDA graphs. While one reviewer noted variance in speedup across tasks, the general consensus is that this comes from inherent task-length differences rather than instability in the proposed method. Given the unanimous positive ratings and the demonstrated acceleration (up to 9.1x), this paper offers a valuable and timely contribution to dLLM community.